# Dissecting the chain of information processing and its interplay with neurochemicals and fluid intelligence across development

George Zacharopoulos[1,2]*, Francesco Sella[1,3], Uzay Emir[1,4], Roi Cohen Kadosh[1,5]*

[1]Wellcome Centre for Integrative Neuroimaging, Department of Experimental Psychology, University of Oxford, Oxford, United Kingdom; [2]School of Psychology, Swansea University, Swansea, United Kingdom; [3]Centre for Mathematical Cognition, Loughborough University, Loughborough, United Kingdom; [4]School of Health Sciences, College of Health and Human Sciences, Purdue University, West Lafayette, United States; [5]School of Psychology, University of Surrey, Guildford, United Kingdom

**Abstract** Previous research has highlighted the role of glutamate and gamma-aminobutyric acid (GABA) in perceptual, cognitive, and motor tasks. However, the exact involvement of these neurochemical mechanisms in the chain of information processing, and across human development, is unclear. In a cross-sectional longitudinal design, we used a computational approach to dissociate cognitive, decision, and visuomotor processing in 293 individuals spanning early childhood to adulthood. We found that glutamate and GABA within the intraparietal sulcus (IPS) explained unique variance in visuomotor processing, with higher glutamate predicting poorer visuomotor processing in younger participants but better visuomotor processing in mature participants, while GABA showed the opposite pattern. These findings, which were neurochemically, neuroanatomically and functionally specific, were replicated ~21 mo later and were generalized in two further different behavioral tasks. Using resting functional MRI, we revealed that the relationship between IPS neurochemicals and visuomotor processing is mediated by functional connectivity in the visuomotor network. We then extended our findings to high-level cognitive behavior by predicting fluid intelligence performance. We present evidence that fluid intelligence performance is explained by IPS GABA and glutamate and is mediated by visuomotor processing. However, this evidence was obtained using an uncorrected alpha and needs to be replicated in future studies. These results provide an integrative biological and psychological mechanistic explanation that links cognitive processes and neurotransmitters across human development and establishes their potential involvement in intelligent behavior.

*For correspondence:
g.zacharopoulos@swansea.ac.uk (GZ);
r.cohenkadosh@surrey.ac.uk (RCK)

## Editor's evaluation

This important study combines behavioral and imaging experiments to understand how levels of important brain chemicals shape the processing of information in the brain in children and young adults. The sample size and data quality are outstanding and some of the data are convincing. However, there are important caveats for some of the results, as discussed by the authors, and future replication will be important to fully substantiate the findings. This work will be of interest to neuroscientists, psychologists, and neuroimaging researchers investigating the developing brain in health and disease.

## Introduction

Previous research has highlighted the role of glutamate and gamma-aminobutyric acid (GABA) in the function and malfunction of the central nervous system underpinning cognitive, decision, and visuo-motor processing (*De la vega et al., 2014*; *Frangou et al., 2019*; *Weidacker et al., 2020*; *Stagg et al., 2011b*; *Cohen Kadosh et al., 2015*; *Zacharopoulos and Cohen Kadosh, 2021b*). However, little is known about the neurochemical mechanisms shaping these processes across development in humans. An influential computational framework has been extensively applied in recent years as it allows the behavioral dissection between cognitive, decision, and visuomotor processes. This framework is based on the drift–diffusion model and analyzes reaction time and accuracy, usually in two-choice response time tasks (*Ratcliff and McKoon, 2008*; *Ratcliff et al., 2016*; *Wagenmakers et al., 2007*).

The diffusion parameters, which we describe in detail in the following paragraph, have shown satisfactory retest reliability and the diffusion paradigm has been widely utilized in a wide range of behavioral tasks assessing various cognitive processes, including attention, lexical decision, recognition memory, priming, and numerical cognition (*Cohen Kadosh et al., 2010*; *van Bueren et al., 2021*; *Huang-Pollock et al., 2012*; *Lerche and Voss, 2017*). Moreover, this robust diffusion framework has been widely applied in non-clinical and clinical domains, including schizophrenia, attention-deficit/hyperactivity disorder (ADHD), visual impairments (*Ratcliff et al., 2016*; *Moustafa et al., 2015*; *Luo et al., 2020*; *Metin et al., 2013*), and in brain-based interventions to target specific mental processes (*van Bueren et al., 2021*). It has been suggested that diffusion models can facilitate the understanding of clinical disorders (*White et al., 2010*). Thus, the application of this model has spanned disparate fields and holds the promise of illuminating the elusive function and malfunction processes of the central nervous system.

While different types of drift–diffusion models exist, one of the models, the EZ-diffusion model (*Wagenmakers et al., 2007*), generates three unobserved variables: mean drift rate (cognitive processing), boundary separation (decision processing), and non-decision time (visuomotor processing). The mean drift rate is typically computed in simple binary choice tasks where humans accumulate evidence in favor of the different alternatives before committing to a decision, reflecting cognitive processing (*Pisauro et al., 2017*). The mean drift rate is a computational construct that assesses the quality of information processes or speed of information uptake processes, where an increase in mean drift rate is believed to induce more accurate and faster decisions (*Zhang and Rowe, 2014*). Boundary separation is a computational construct that assesses response conservativeness regarding the decision criterion. The trade-off between decision speed and accuracy is thought to be created by changing the boundary separation (*Zhang and Rowe, 2014*). Non-decision time is a computational construct that assesses the subset of the reaction time devoted to perceptual processes and motor response execution. Namely, non-decision time is the time spent on processing other than the decision process, and it is usually referred to as reflecting the early perceptual processing of the stimulus of interest and the implementation of the motor response once the decision process is completed, reflecting visuomotor processing (*Ratcliff and McKoon, 2008*; *Philiastides and Ratcliff, 2013*; *Mulder et al., 2012*). Similar to mean drift rate and boundary separation, non-decision time exhibits well-characterized developmental shifts. Children exhibited lower mean drift rate, higher boundary separation, and longer non-decision time compared to college-aged adults in numerical discrimination tasks (*Ratcliff et al., 2012*; *von Krause et al., 2020*).

*Which brain regions are involved in the diffusion parameters?* Frontal and parietal regions are thought to track evidence accumulation before reaching a decision in a task-independent way (*Pisauro et al., 2017*; *Ratcliff et al., 2012*; *Scott et al., 2017*). Moreover, a study combining functional MRI fMRI and the drift–diffusion model identified that frontoparietal areas corresponded to the decision variables resulting from the downstream stimulus–criterion comparison independent of stimulus type (*White et al., 2012*).

The involvement of mainly frontal and parietal regions in encoding the diffusion parameters raises the possibility that neurodevelopmental changes within frontoparietal network regions may shape the well-documented developmental fluctuations in diffusion parameters, especially in tasks that mainly recruit the frontoparietal network, such as attention and numerical processing.

In a similar vein, a recent magnetic resonance spectroscopy (MRS) study revealed that glutamate and GABA profiles within the classic frontoparietal network regions, namely, the intraparietal sulcus

(IPS, see *Figure 1—figure supplement 2A–C*) and the middle frontal gyrus (MFG, see *Figure 1—figure supplement 2D–F*), explained current and predicted future mathematical achievement levels (*Zacharopoulos et al., 2021c*).

Given the well-established developmental fluctuations in the diffusion parameters over the life span, it has been proposed that developmental shifts in all three diffusion parameters are possible explanations for the corresponding developmental shifts in cognitive performance (*von Krause et al., 2020*). For example, several studies found that individual differences in intelligence are primarily related to the mean drift rate (*Schmiedek et al., 2007*; *Schubert and Frischkorn, 2020*; *Schulz-Zhecheva et al., 2016*). In a recent study on aging (ages 18–62) using 18 different response time tasks, age differences in intelligence were accounted for age differences in non-decision time (*von Krause et al., 2020*). If this latter relationship also extends to earlier developmental windows, this raises the intriguing possibility that the link between the frontoparietal regions and cognition across development is at the visuomotor level (i.e., non-decision time), rather than the more cognitive level (i.e., mean drift rate). To assess these open questions, we utilized an attention network task (*Figure 1*, Task 1) because (i) it relies mainly on the frontal and parietal regions, (ii) it is suited for the calculation of the three diffusion parameters, and (iii) its performance is sensitive to developmental changes (*Petersen and Posner, 2012*). We focused primarily on glutamate and GABA as these neurotransmitters were shown to be involved in attention and attention-related pathologies, including ADHD, in both human and animal work (*Pehrson et al., 2013*; *Edden et al., 2012*; *Moore et al., 2006*; *Rüsch et al., 2010*; *Yang et al., 2010*; *Aoki et al., 2013*; for extended text and references, see 'Materials and methods' section).

Given the critical role of the frontal and parietal regions, the role of development, and the role of neurochemicals in perceptual, cognitive, and motor tasks, several key research questions emerge: (i) How is frontoparietal neurochemical concentration related to mean drift rate, boundary separation, and non-decision time (henceforth, we will refer to non-decision time as visuomotor processing), and how do these associations change across human development? (ii) Do these associations between frontoparietal neurochemicals and visuomotor processing exist in a task-independent manner? In other words, do the associations between frontoparietal neurochemicals and visuomotor processing replicate across different tasks such as numerical processing (*Figure 1*, Task 2) and mental rotation (*Figure 1*, Task 3)? (iii) Do the associations between frontoparietal neurochemicals and visuomotor processing explain individual differences in fluid intelligence?

An informative way to address these questions is by using MRS. Several advantages of this technology make it suitable for studying these processes in the context of development. MRS has shown great promise in the classification and prediction of certain medical conditions in prior investigations. Therefore, it is possible to utilize and extend this unique predictive ability of MRS to discern which of these key neurobiological processes (cognitive, decision, visuomotor) are associated with highly specialized neurochemical concentrations across development (*Porges et al., 2021*). As mentioned above, our main aims were to examine whether neurochemical concentration within key frontoparietal regions can explain cognitive (mean drift rate), decision (boundary separation), and visuomotor processing. By testing participants from early childhood to early adulthood, we were able to examine whether these associations are static or dynamic across the life span.

## Results
### Task 1
### Developmental trajectories of the mean drift rate, boundary separation, and visuomotor processing

As a first step, we tracked the developmental trajectories of each of the three diffusion parameters (mean drift rate, boundary separation, visuomotor processing) during the first and the second assessment (for scatterplots, see *Supplementary file 2*; for statistical values, see *Supplementary file 4*). Lower scores in boundary separation and visuomotor processing indicate better performance, whereas higher scores in mean drift rate indicate better performance. As shown in *Supplementary file 4*, increasing age was associated with lower boundary separation and visuomotor processing and a higher mean drift rate, as expected. All these effects were replicated independently during the second assessment (*Supplementary file 4*).

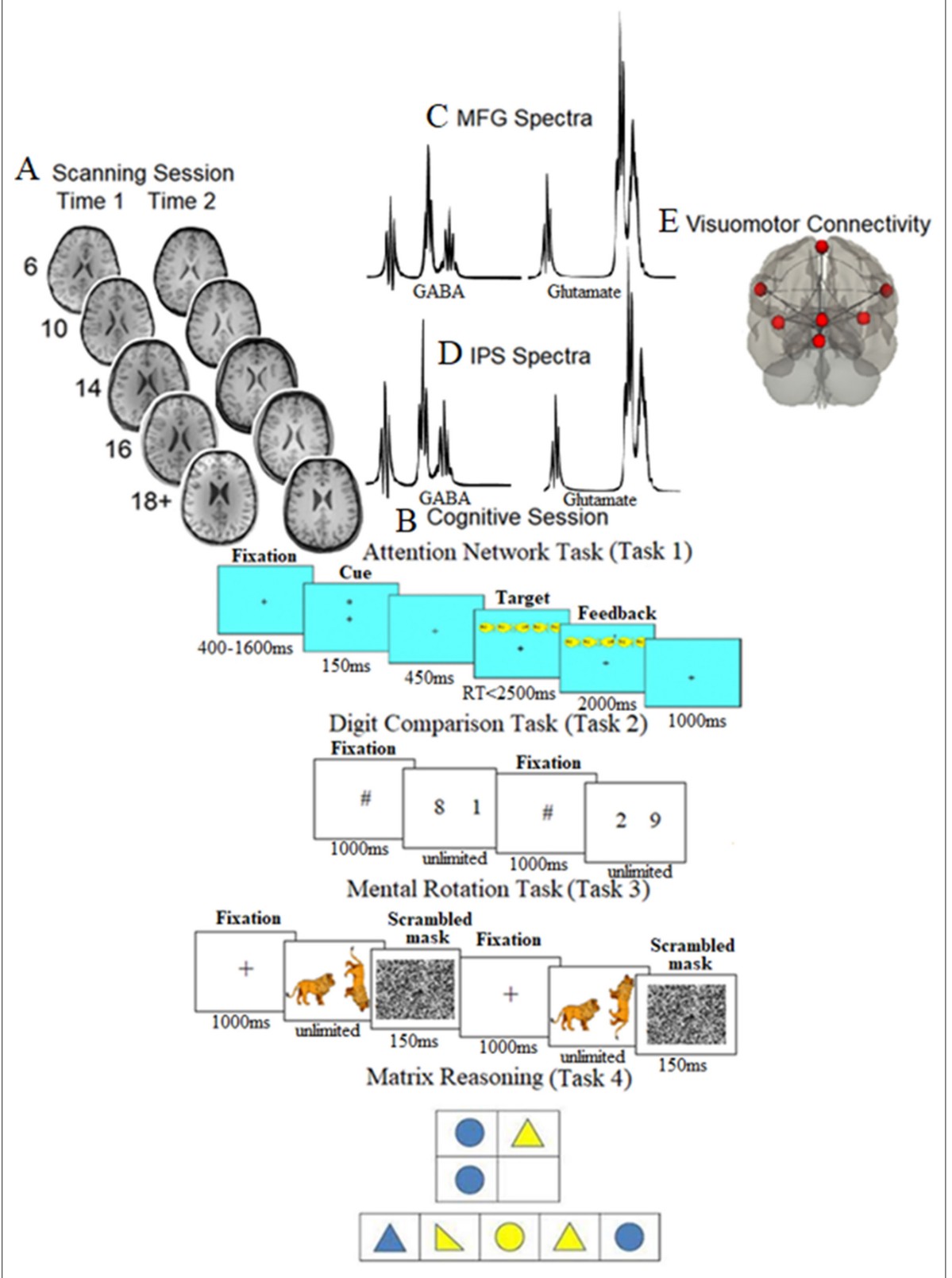

**Figure 1.** Scanning, cognitive session, neurochemical spectra plots, and visuomotor network connectivity. (**A**) Examples of T1-weighted images collected during the first assessment (A1) and the second assessment (A2, ~ 21 mo later) in each of the five age groups (at A1: 6-year-olds, 10-year-olds, 14-year-olds, 16-year-olds, and 18+-year-olds). (**B**) Cognitive session: *top panel*: the Attention Network task, where after a variable period, one of the possible cue types was presented for 150 ms; after a pause of 450 ms, one of the possible target types was presented, which required participants

*Figure 1 continued on next page*

*Figure 1 continued*

to indicate the direction of the middle fish arrow by pressing the corresponding key; *top middle panel*: the Digit Comparison task, where participants chose the larger value between two single-digit Arabic numbers. The numbers remained on the screen until the participant responded. Between each trial, a central fixation hashtag appeared for 1000 ms. *Bottom middle panel*: the Mental Rotation task, where participants decided whether a rotated target figure (a giraffe, a zebra, or a racoon in the actual task) presented on the right side of the screen was the same (i.e., non-mirrored) or different (i.e., mirrored) compared to the upright figure presented on the left side of the screen. Each trial began with a black fixation cross in the center of the white screen for 1000 ms. *Bottom panel*: matrix reasoning, which is a 30-item (of increasing difficulty) assessment tool that requires identifying a logical pattern in a sequence of visuospatial stimuli (this panel presents an example for illustration purposes only and does not show a trial from WASI II; *Wechsler, 2011*). (**C, D**) The mean spectra from our sample for the middle frontal gyrus (MFG) (**C**) and intraparietal sulcus (IPS) (**D**). The thickness corresponds to ±1 SD from the mean (i.e., a chemical shift expressed in parts per million, ppm, on the x-axis). (**E**) Brain regions consisting of the resting-state visuomotor functional connectivity network.

The online version of this article includes the following figure supplement(s) for figure 1:

**Figure supplement 1.** A stacked plot depicting the spectrum, the fit, the residual, the baseline estimation, and the individual contributions from all basis functions.

**Figure supplement 2.** The positions displayed in a T1-weighted image for the intraparietal sulcus (IPS) and the middle frontal gyrus (MFG) on coronal, sagittal, and axial slices.

## Neurochemicals and behavioral performance

After establishing the developmental trajectories of the three behavioral scores, we examined the capacity of neurochemicals to track these behavioral scores across development. For brevity, we present the neurochemical results that survived FDR correction and were neurochemically, neuroanatomically, and diffusion-parameter specific (i.e., the results from assessing a given diffusion parameter hold even after controlling for the other two diffusion parameters) and replicated during the second assessment (see 'Materials and methods' section for a complete description of this selection process).

There was a significant interaction between age and neurochemical measures in tracking visuomotor processing, which was replicated in the second assessment: IPS glutamate * age (first assessment: *Figure 2A*, $\beta$ = –0.22, t(252) = –5.38, $P_{BO}$< 0.001, CI = [–0.31, –0.11]; second assessment: *Figure 2C*, $\beta$ = –0.24, t(174) = –4.47, $P_{BO}$ < 0.001, CI = [–0.35, –0.11]) and IPS GABA * age (first assessment: *Figure 2B*, $\beta$ = 0.23, t(252) = 5.71, $P_{BO}$ < 0.001, CI = [0.14, 0.32]; second assessment: *Figure 2D*, $\beta$ = 0.24, t(175) = 4.41, $P_{BO}$ = 0.002, CI = [0.11, 0.41]). Specifically, high glutamate predicted shorter visuomotor processing time in mature participants and longer visuomotor processing time in younger participants while GABA showed the opposite pattern. Additional analyses presented in *Supplementary file 15* revealed that there were independent relationships between IPS glutamate * age and IPS GABA * age with task performance, as both predictors (i.e., IPS glutamate * age and IPS GABA * age) explained unique variance in task performance (visuomotor processing) even when both were added in the same multiple regression model.

## Task 2

The results of Task 1 suggested that the neurochemical profile of individuals tracked individual variation in visuomotor processing across development and that these associations were neurochemically, neuroanatomically, and diffusion-parameter specific to visuomotor processing. The aim of Task 2 was to replicate this set of findings that were obtained using the attention network task (Task 1) with a different task: a digit comparison task (see *Figure 1B* and the 'Materials and methods' section).

### Developmental trajectories of the mean drift rate, boundary separation, and visuomotor processing

The behavioral results of Task 2 mirrored the results of Task 1: increasing age was associated with lower boundary separation and visuomotor processing and with a higher mean drift rate (*Supplementary file 4*; for scatterplots, see, *Supplementary file 2*).

### Neurochemicals and behavioral performance

As in Task 1, we found a significant interaction between age and neurochemical measures in tracking visuomotor processing in Task 2: IPS glutamate * age (first assessment: *Figure 3A*, $\beta$ = –0.17, t(240) = –4.25, $P_{BO}$< 0.001, CI=[–0.26, –0.09]; second assessment: *Figure 3C*, $\beta$ = –0.16, t(170) = –3.17, $P_{BO}$ =

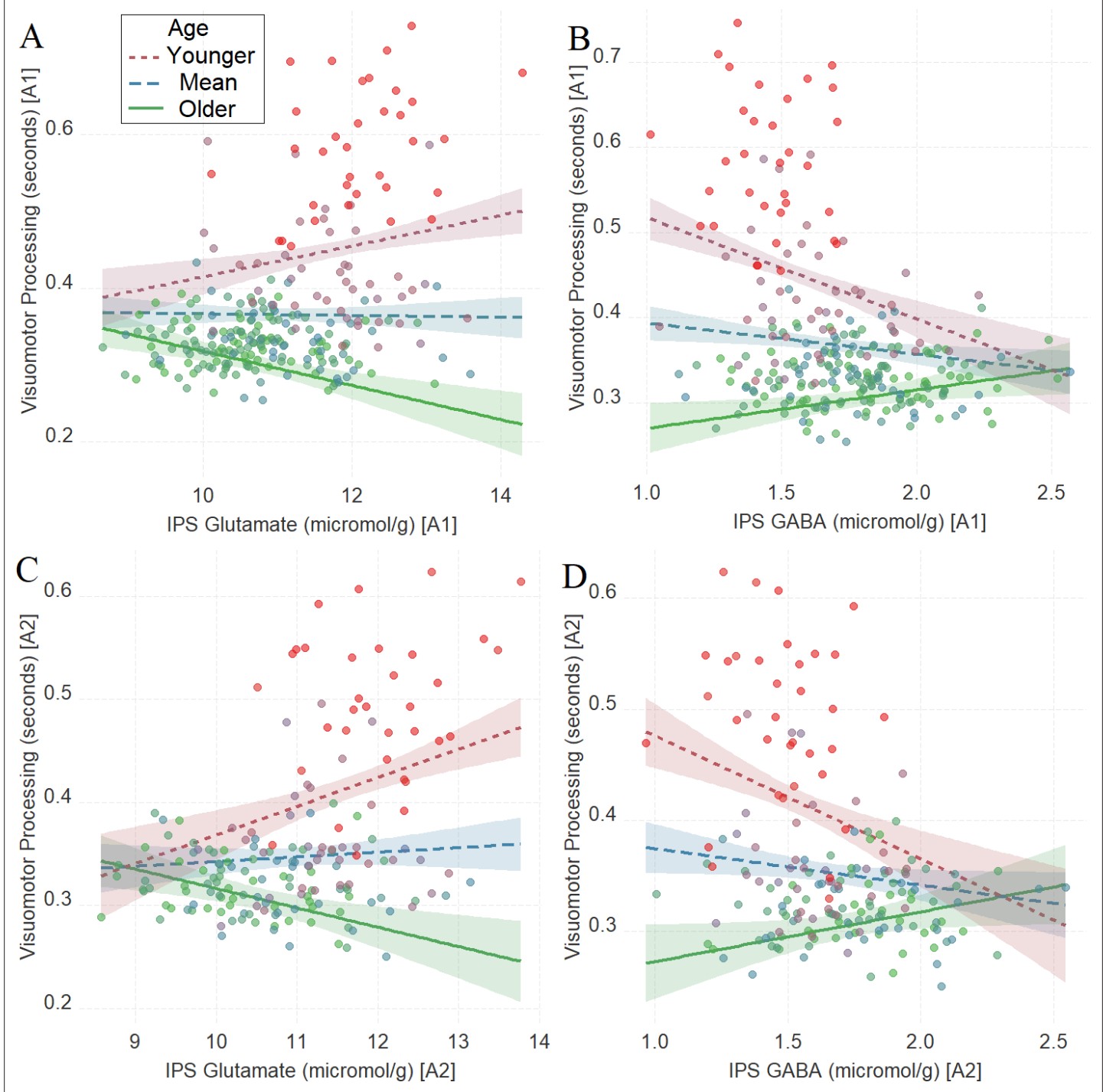

**Figure 2.** The moderating role of age in the relation between neurotransmitter concentration and visuomotor processing at A1 (**A, B**) and A2 (**C, D**) in Task 1. (**A**) Intraparietal sulcus (IPS) glutamate * age and (**B**) IPS GABA * age at A1; (**C**) IPS glutamate * age and (**D**) IPS GABA * age at A2. We utilized a standard practice for examining interactions with continuous variables using three levels of a moderator (*Aiken et al., 1991*). Specifically, to depict the interaction between the continuous variables (age and neurotransmitter concentration), we plotted the regression lines for younger and older participants (i.e., ±1 SD) from the mean age (*Aiken et al., 1991*). Green colors represent +1 SD above the mean (older participants), red colors represent −1 SD below the mean (younger participants), and blue colors represent the mean. Since our cohort was based on a developmental sample spanning from early childhood to early adulthood, 'younger participants' refer predominantly to early and late childhood and 'older participants' refer predominantly to late adolescence and early adulthood. To make the visual comparison across figures more comparable, we did not control for boundary separation and mean drift rate when plotting these panels. We additionally plotted the moderating role of age in the relation between neurotransmitter concentration and visuomotor processing after controlling for boundary separation and mean drift rate in *Figure 2—figure*

*Figure 2 continued on next page*

*Figure 2 continued*

*supplement 1*. Lower scores in visuomotor processing indicate better performance. The shaded area represents 95% confidence intervals. In these multiple regression analyses, the sample sizes were 258, 180, 258, and 181, for panels A, B, C and D respectively, and the significance level was set to 0.05 (5%).

The online version of this article includes the following figure supplement(s) for figure 2:

**Figure supplement 1.** Plotting the moderating role of age in the relation between neurotransmitter concentration and visuomotor processing after controlling for boundary separation and mean drift rate in Task 1 (glutamate: **A**; GABA: **B**), Task 2 (glutamate: **C**; GABA: **D**), and Task 3 (glutamate: **E**; GABA: **F**).

**Figure supplement 2.** Scatterplots depicting that age moderated the relationship between the (z-scored) diffusion parameter (**A**: mean drift rate, **B**: boundary separation, **C**: visuomotor processing) and fluid intelligence.

---

0.017, CI = [−0.28, −0.02]) and IPS GABA * age (first assessment: *Figure 3B*, $\beta$ = 0.18, t(240) = 5.10, $P_{BO}$< 0.001, CI = [0.08, 0.27]; second assessment: *Figure 3D*, $\beta$ = 0.29, t(168) = 6.02, $P_{BO}$ < 0.001, CI = [0.18, 0.4]).

## Task 3

The results of Tasks 1–2 combined suggested that the neurochemical profile of individuals tracked individual variation in visuomotor processing across development. In Task 3, we further examined this set of findings in a yet different task: a mental rotation task (see *Figure 1B* and the 'Materials and methods' section).

### Developmental trajectories of the mean drift rate, boundary separation, and visuomotor processing

The behavioral results of Task 3 reflected the results of Tasks 1–2 in that increasing age was associated with lower boundary separation and visuomotor processing and a higher mean drift rate (*Supplementary file 4*; for scatterplots, see *Supplementary file 2*).

### Neurochemicals and behavioral performance

Similar to Tasks 1–2, we found a significant interaction between age and neurochemical measures in tracking visuomotor processing: IPS glutamate * age (first assessment: *Figure 4A*, $\beta$ = −0.27, t(222) = −5.02, $P_{BO}$ < 0.001, CI = [−0.42, −0.11], which was not significant in the second assessment: *Figure 4C*, $\beta$ = 0.11, t(164) = 1.67, $P_{BO}$ = 0.21, CI = [−0.05, 0.3]) and IPS GABA * age (first assessment: *Figure 4B*, $\beta$ = 0.27, t(224) = 4.66, $P_{BO}$ = 0.001, CI = [0.11, 0.43]; second assessment: *Figure 4D*, $\beta$ = 0.39, t(166) = 5.46, $P_{BO}$ < 0.005, CI = [0.22, 0.56]).

## Neurochemicals, visuomotor connectivity, and behavioral performance

After establishing the impact of IPS glutamate and GABA on tracking visuomotor processing in the same way across the three tasks as a function of development, we examined whether this relationship is influenced by visuomotor connectivity (see the 'Materials and methods' section for further details on the composite visuomotor processing score, which reflects visuomotor processing across the three tasks). We found that visuomotor connectivity interacted with age in tracking visuomotor processing (*Figure 5C*, $\beta$ = −0.16, t(215) = −3.9, $P_{BO}$ = 0.002, CI = [−0.26, −0.06]), and we also found that IPS glutamate (*Figure 5A*, $\beta$ = −0.18, t(242) = −2.64, $P_{BO}$ = 0.022, CI = [−0.33, −0.02]) and IPS GABA (*Figure 5B*, $\beta$ = 0.15, t(243) = 2.10, $P_{BO}$ = 0.017, CI = [0.02, 0.26]) interacted with age in tracking visuomotor connectivity.

This set of findings raises the possibility that IPS neurochemicals (i.e., glutamate and/or GABA) shape visuomotor processing via visuomotor connectivity across development. To assess this possibility, we ran a moderated mediation (model 59; see 'Materials and methods' section for details) where we evaluated whether the neurotransmitter concentration (independent variable) tracks visuomotor processing (dependent variable) via the visuomotor connectivity (mediator) as a function of development (moderator).

We found that the involvement of GABA and glutamate depends on the developmental stage: the relationship between IPS GABA and visuomotor processing is mediated by visuomotor connectivity only for the younger participants (indirect effect = −0.03, CI = [−0.06, −0.001]; for full details, see

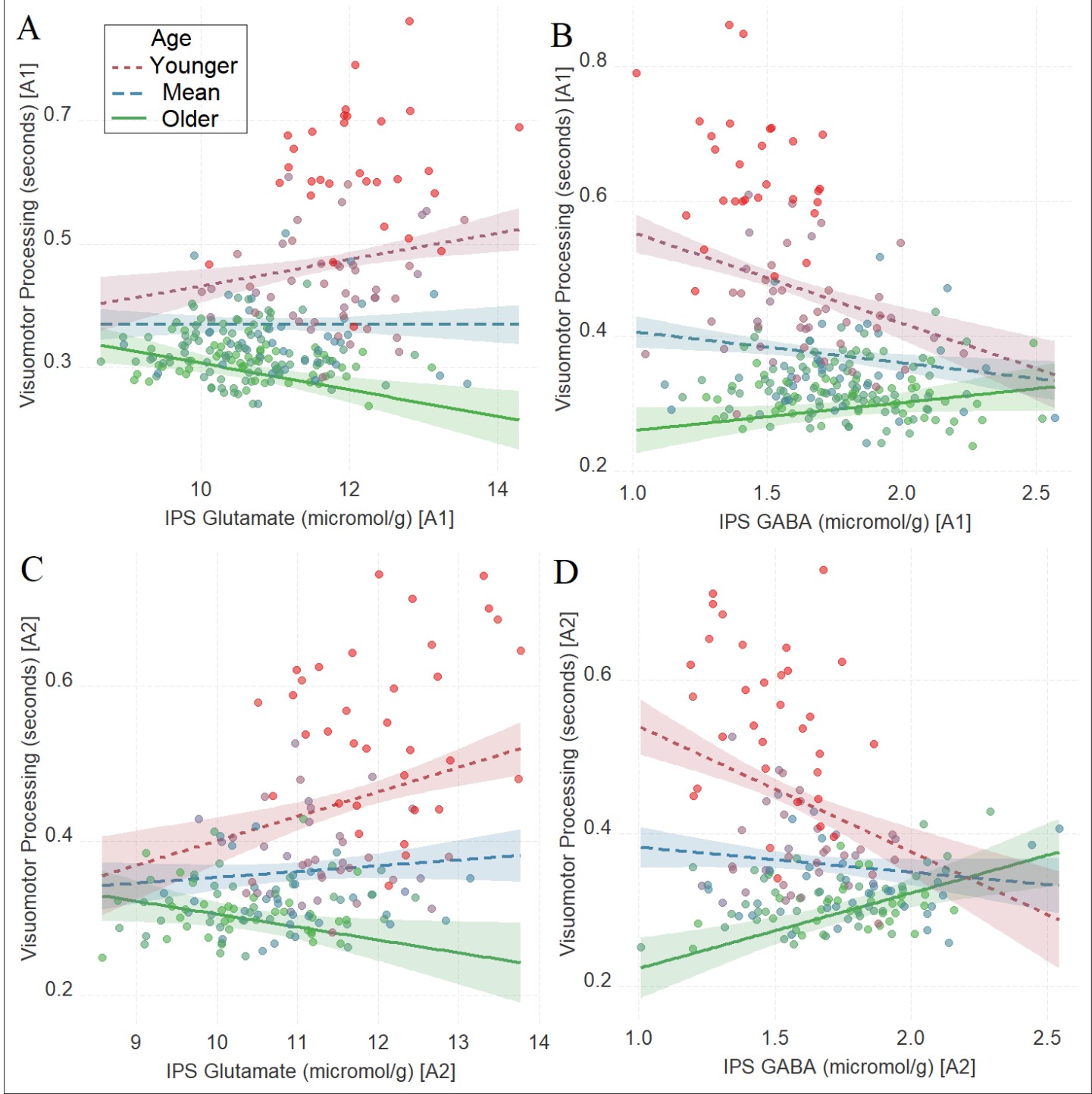

**Figure 3.** The moderating role of age in the relation between neurotransmitter concentration and visuomotor processing at A1 (**A, B**) and A2 (**C, D**) in Task 2. (**A**) Intraparietal sulcus (IPS) glutamate * age and (**B**). IPS GABA * age at A1; (**C**) IPS glutamate * age and (**D**) IPS GABA * age at A2. We utilized a standard practice for examining interactions with continuous variables using three levels of a moderator (*Aiken et al., 1991*). Specifically, to depict the interaction between the continuous variables (age and neurotransmitter concentration), we plotted the regression lines for younger and older participants (i.e., ±1 SD) from the mean age (*Aiken et al., 1991*). Green colors represent +1 SD above the mean (older participants), red colors represent –1 SD below the mean (younger participants), and blue colors represent the mean. Since our cohort was based on a developmental sample spanning from early childhood to early adulthood, 'younger participants' refer predominantly to early and late childhood and 'older participants' refer predominantly to late adolescence and early adulthood. To make the visual comparison across figures more comparable, we did not control for boundary separation and mean drift rate when plotting these panels. Lower scores in visuomotor processing indicate better performance. The shaded

*Figure 3 continued on next page*

*Figure 3 continued*

area represents 95% confidence intervals. In these multiple regression analyses, the sample sizes were 246, 176, 246, and 174, for panels A, B, C and D respectively, and the significance level was set to 0.05 (5%).

*Supplementary file 7.1*). By contrast, visuomotor connectivity showed a trend in mediating the relationship between IPS glutamate and visuomotor processing only for the older participants (indirect effect = 0.02, CI = [–0.003, 0.06]; for full details, see *Supplementary file 7.2*).

### Neurochemicals and intelligence scores

As mentioned in the 'Introduction,' a behavioral study found that age differences in intelligence were accounted for by age differences in visuomotor processing (*von Krause et al., 2020*). Importantly, the IPS has been highlighted as one of the brain hubs for fluid intelligence (*Jung and Haier, 2007*). Since we identified a reliable task-independent developmental effect between IPS neurochemicals and visuomotor processing, we utilized Task 4 to assess whether visuomotor processing mediates the relationship between neurochemicals and fluid intelligence (see *Figure 1B* and the 'Materials and methods' section).

As in Tasks 1–3, there was a significant interaction between age and glutamate in tracking intelligence scores: IPS glutamate * age (first assessment: *Figure 6A*, $\beta$ = 0.16, t(256) = 3.72, $P_{BO}$ < 0.001, CI = [0.06, 0.25]; second assessment: *Figure 6C*, $\beta$ = 0.12, t(178) = 2.09, $P_{BO}$ = 0.02, CI = [0.02, 0.23]). There was a significant interaction between age and GABA in tracking intelligence scores during the second assessment: IPS GABA * age (first assessment: *Figure 6B*, $\beta$ = –0.05, t(257) = –1.09, $P_{BO}$ = 0.27, CI = [–0.13, 0.04]; second assessment: *Figure 6D*, $\beta$ = –0.15, t(178) = –2.40, $P_{BO}$ = 0.01, CI = [–0.25, –0.05]).

This set of findings raises the possibility that IPS neurochemicals shape fluid intelligence via visuomotor processing across development. To assess this possibility, we ran a moderated mediation (model 59; see 'Materials and methods' section for details) where we evaluated whether the neurotransmitter concentration tracking of fluid intelligence as a function of age is mediated by visuomotor processing.

Visuomotor processing mediated the relationship between IPS GABA and fluid intelligence only for the older participants (*Figure 7A*, indirect effect = 0.04, CI = [0.002, 0.1]; for full details, see *Supplementary file 7.3*). Similarly, visuomotor processing mediated the relationship between IPS glutamate and fluid intelligence only for the older participants (*Figure 7B*, indirect effect = –0.06, CI = [–0.12, –0.006]; for full details, see *Supplementary file 7.4*).

## Discussion

This study examined the neurochemical mechanisms underlying cognitive, decision, and visuomotor processing across development by focusing on glutamate and GABA within the frontoparietal network. Four main findings emerged from our study: (i) visuomotor processing, as assessed with the computational metric non-decision time, is associated with glutamate and GABA within the IPS in a developmentally dependent manner, and was replicated in two additional tasks involving non-identical cognitive processing; (ii) the link between visuomotor resting-state connectivity and visuomotor processing, as well as IPS glutamate and GABA, is developmentally dependent; (iii) visuomotor connectivity mediates the relationship between IPS neurochemicals and visuomotor processing; and (iv) visuomotor processing mediates the relationship between IPS neurochemicals and fluid intelligence depending on the developmental stage.

### Visuomotor processing is associated with IPS glutamate and GABA in a developmentally dependent manner

Our results reveal how IPS glutamate and GABA, the neurotransmitters involved in brain excitation and inhibition (*Barron et al., 2016*; *Hone-Blanchet et al., 2016*; *Kim et al., 2014*; *Stagg et al., 2009*), are associated with visuomotor functions in a developmentally dependent manner. Across three cognitive tasks, we showed that high glutamate predicted shorter visuomotor processing time in mature participants and longer visuomotor processing time in younger participants, while GABA showed the opposite pattern. These findings suggest that the relationship between IPS glutamate and GABA and visuomotor processing across development is task-independent.

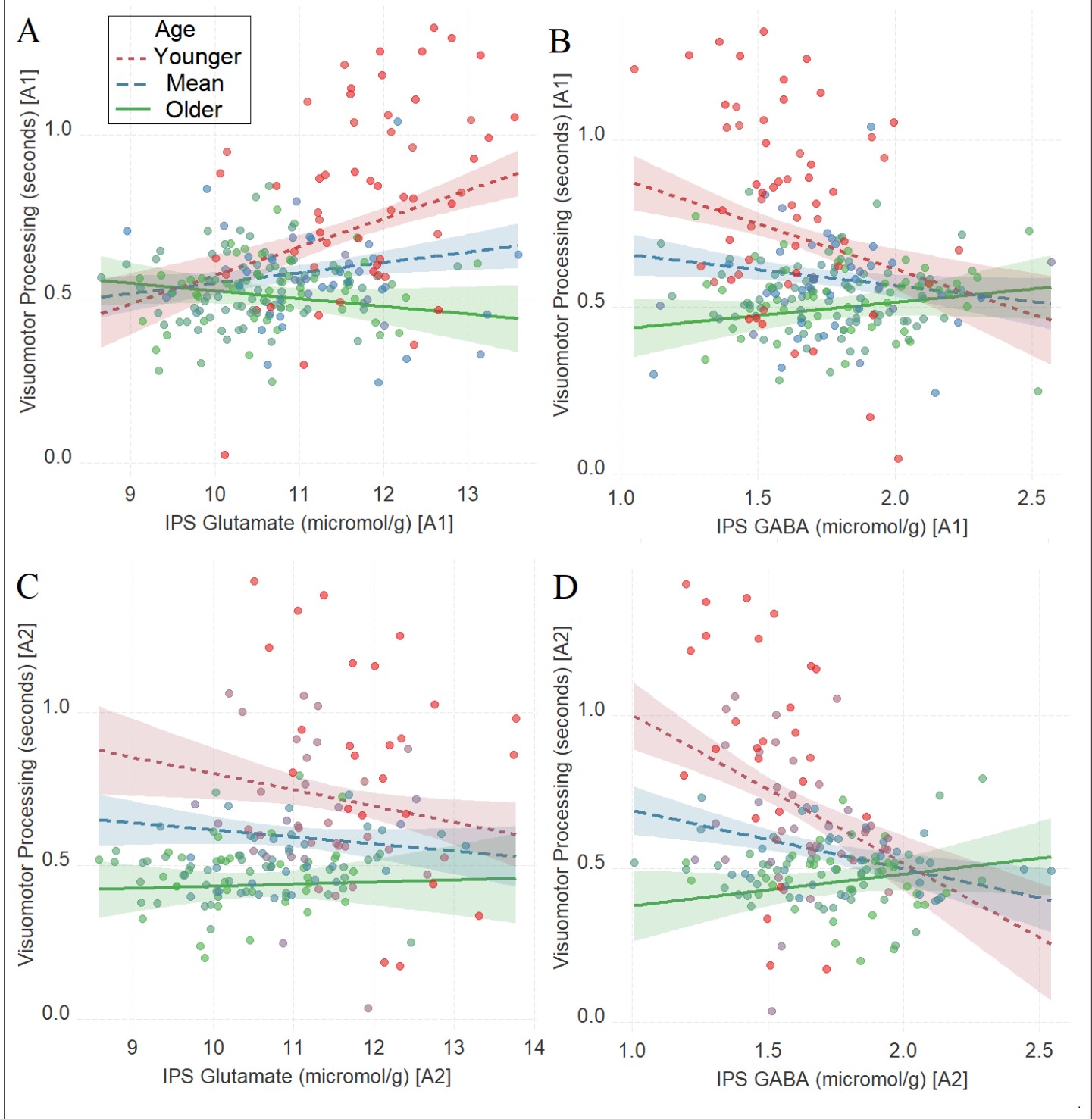

**Figure 4.** The moderating role of age in the relation between neurotransmitter concentration and visuomotor processing at A1 (**A, B**) and A2 (**C, D**) in Task 3. (**A**) Intraparietal sulcus (IPS) glutamate * age and (**B**) IPS GABA * age at A1; (**C**) IPS glutamate * age and (**D**) IPS GABA * age at A2. We utilized a standard practice for examining interactions with continuous variables using three levels of a moderator (*Aiken et al., 1991*). Specifically, to depict the interaction between the continuous variables (age and neurotransmitter concentration), we plotted the regression lines for younger and older participants (i.e., ±1 SD) from the mean age (*Aiken et al., 1991*). Green colors represent +1 SD above the mean (older participants), red colors represent −1 SD below the mean (younger participants), and blue colors represent the mean. Since our cohort was based on a developmental sample spanning from early childhood to early adulthood, 'younger participants' refer predominantly to early and late childhood and 'older participants' refer predominantly to late adolescence and early adulthood. To make the visual comparison across figures more comparable, we did not control for

*Figure 4 continued on next page*

*Figure 4 continued*

boundary separation and mean drift rate when plotting these panels. Lower scores in visuomotor processing indicate better performance. The shaded area represents 95% confidence intervals. In these multiple regression analyses, the sample sizes were 228, 170, 230, and 172, for panels A, B, C and D respectively, and the significance level was set to 0.05 (5%).

Even though not clearly established yet, the age trajectories of GABA and glutamate during childhood and early adulthood have been examined in several studies to date. For example, there is evidence for increasing GABA and macromolecules during childhood (*Porges et al., 2021*), but this has been ascribed to macromolecules rather than GABA itself (*Bell et al., 2021*). In a separate investigation using this study's sample, we have found increases in GABA and decreases in glutamate from early childhood to early adulthood (for extensive discussion on the age trajectories of MRS-based metabolites literature, see *Zacharopoulos et al., 2021a*). Importantly, previous studies have examined the role of developmentally dependent glutamate levels in cognitive, emotional, and, importantly, motor functions (*Cobo and Mora, 1991*; *Collingridge and Bliss, 1987*; *Cotman et al., 1988*; *Mora and Cobo, 1991*; *Nicolle et al., 1996*; *Nicolle and Baxter, 2003*; *Schmidt et al., 1992*; *Segovia et al., 2001*; *Zahr et al., 2008*). However, these studies focused primarily on adulthood and suggested that alterations in the availability of biochemical glutamate might contribute to the neural mechanisms underlying age-related cognitive and motor impairments (*Zahr et al., 2013*). Our findings that in later developmental stages higher levels of IPS glutamate are associated with better visuomotor performance (*Figures 2A and C, 3A and C, and 4A and C*) are overall in line with the previous studies in older adults and extend them (*Zahr et al., 2013*). Importantly, we obtained a negative relationship in the early developmental stages, where higher levels of IPS glutamate were associated with poorer visuomotor performance (*Figures 2A and C, 3A and C, and 4A and C*). A possible explanation, as we also mention below, is that these glutamate-induced visuomotor transformations within the human IPS from early childhood to early adulthood may specifically occur within the IPS neurons, identified both in the human and in the non-human primate brain, that respond to the visuomotor domain (*Grefkes and Fink, 2005*). The accuracy of this working hypothesis can be tested in future animal pharmacological studies capable of reaching the necessary neuronal resolution that was beyond the reach of this study using analogous tasks. Although the measurements from both the MFG and the IPS yielded high spectral quality, due to degraded shimming conditions in MFG, the line width of MFG was slightly higher (MFG = 0.027, IPS = 0.022, p<0.001). Due to the excellent spectral quality, we think that these differences are unlikely to explain the lack of effect of MFG GABA and glutamate, but the reader should be aware of such a possibility. This study was conceptualized and conducted based on the assumptions of a linear relationship between diffusion parameters and age. However, as one of the reviewers has highlighted, more complex nonlinear relationships may exist between diffusion parameters and age. We, therefore, present in *Supplementary file 4* linear quadratic and cubic fits between between diffusion parameters and age (see also, *Supplementary file 16*), and future studies can be conceptualized and optimized for assessing nonlinear associations.

Regarding GABA, the maturation of GABA circuits and in particular the maturation of parvalbumin cells, a positive subtype of GABA neurons, is thought to be one of the molecular signatures triggering the onset of sensitive periods and plasticity, where an experimental increase or reduction of GABA triggers a precocious or delayed onset of sensitive periods/plasticity, respectively (*Zacharopoulos et al., 2021c*; *Werker and Hensch, 2015*). One possibility, which we had suggested previously in the context of high-level cognition and we extended here to visuomotor processing (*Zacharopoulos et al., 2021c*), is that elevated GABA in early development may indicate greater plasticity, leading both to better visuomotor processing (i.e., shorter non-decision time) and less positive connectivity in the visuomotor network. Regarding the relation between GABA levels and visuomotor processing in mature individuals, our findings indicate the opposite pattern of associations: elevated GABA leads both to poorer visuomotor processing (i.e., longer non-decision time) and more positive connectivity in the visuomotor network. Previous studies on GABA and cognition in adults, using modest sample sizes, yielded some conflicting results in that some studies found reduced GABA levels to be associated with cognitive improvement while others found the opposite pattern. For example, reduced GABA was associated with learning improvement in the motor system (*Kolasinski et al., 2019*; *Stagg et al., 2011a*; *Floyer Lea et al., 2006*) and the visual system (*Lunghi et al., 2015*; *Frangou et al., 2019*), although some types of visual learning, and sensory learning in the tactile system, were associated

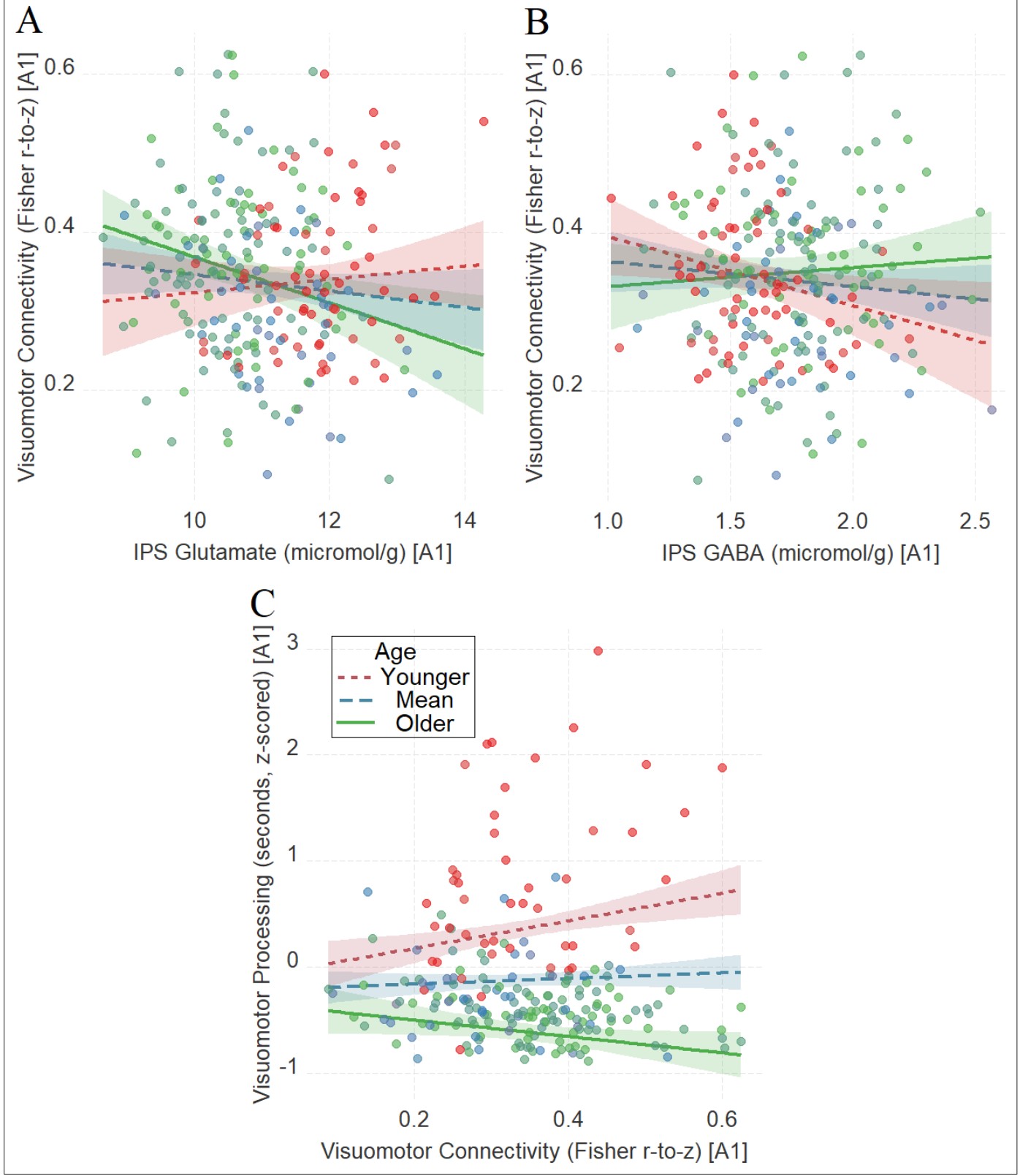

**Figure 5.** The moderating role of age in the relation between neurochemical concentration and visuomotor connectivity in intraparietal sulcus (IPS) glutamate (**A**) and GABA (**B**), and the moderating role of age in the relation between visuomotor connectivity and the composite visuomotor processing score (**C**). We utilized a standard practice for examining interactions with continuous variables using three levels of a moderator (*Aiken et al., 1991*). Specifically, to depict the interaction between the continuous variables (age and neurotransmitter concentration), we plotted the regression lines for

*Figure 5 continued on next page*

*Figure 5 continued*
younger and older participants (i.e., ±1 SD) from the mean age (*Aiken et al., 1991*). Green colors represent +1 SD above the mean (older participants), red colors represent −1 SD below the mean (younger participants), and blue colors represent the mean. Since our cohort was based on a developmental sample spanning from early childhood to early adulthood, 'younger participants' refer predominantly to early and late childhood and 'older participants' refer predominantly to late adolescence and early adulthood. Lower scores in visuomotor processing indicate better performance. The shaded area represents 95% confidence intervals. In these multiple regression analyses, the sample sizes were 246, 247, and 219, for panels A, B, and C respectively, and the significance level was set to 0.05 (5%).

with increased GABA (*Frangou et al., 2019*; *Frangou et al., 2018*; *Heba et al., 2016*; *Shibata et al., 2017*). There are several reasons for these apparent discrepancies. For example, it is important to mention that the target MRS brain regions vary between these studies, and one cortical area may not represent or generalize to other cortical areas in terms of the association of neurotransmitter levels with the process under investigation. Besides these discrepancies, our findings in the mature participants suggest that lower GABA concentration within the IPS is associated with enhanced visuomotor processing, thus extending the involvement of GABA in the acquisition of visuomotor abilities. In addition to our suggestion of the involvement of GABA in visuomotor processing during development, administration of GABA_B agonist baclofen was shown to impair visuomotor processing in healthy adults (*Johnstone et al., 2021*). Consistent with this, we showed here that elevated GABA levels in the IPS are associated with poor visuomotor processing in later developmental stages. Importantly, because we employed a cross-sectional design, we were able to show that the relationship between GABA and visuomotor processing changes between early childhood and early adulthood. Although our sample included healthy individuals with a varied range of abilities, this finding can have implications for methods and applications aimed at detecting developmental atypicalities, and at altering GABA levels in earlier developmental stages in the case of visuomotor disorders.

The obtained results highlight the involvement of the IPS, but not the MFG, in visuomotor processing. These findings are in line with previous studies that have shown that visuomotor processing in the macaque and human IPS supports goal-directed behavior in a modality-general manner (*Grefkes et al., 2004*; *Culham et al., 2003*; *Frey et al., 2005*; *Prado et al., 2005*). Our findings support and extend this observation by identifying specific neurobiological markers within the IPS (i.e., glutamate and GABA) that are associated with goal-directed behavior from childhood to adulthood across four tasks. Moreover, our findings extend the results of previous studies, which were conducted using several modalities (*Stagg et al., 2009*; *Kolasinski et al., 2019*; *Stagg et al., 2011a*; *Floyer Lea et al., 2006*; *Johnstone et al., 2021*; *Bachtiar et al., 2018*; *O'Shea et al., 2017*; *Petitet et al., 2018*), by showing that the involvement of GABA in visuomotor processing extends beyond the sensorimotor cortex and into the IPS.

## Visuomotor connectivity mediates the relationship between IPS neurochemicals and visuomotor processing

To delve deeper into the mechanistic network level, we employed resting fMRI and calculated the within-network connectivity of the visuomotor network for each participant. This visuomotor connectivity was associated with visuomotor processing and with IPS neurochemicals, providing a novel mechanistic account by which IPS neurochemicals, visuomotor networks, and development may interact to track the visuomotor functions of the human brain. Our findings support previous accounts suggesting that IPS serves as an interface between the sensory and motor systems (*Grefkes et al., 2004*), thereby revealing a developmentally dynamic and dissociable function of glutamate and GABA in shaping visuomotor processing that is replicable in multiple cognitive tasks.

The multimodal moderated mediation analyses revealed that IPS GABA and glutamate track visuomotor processing, as reflected by non-decision time scores, via visuomotor connectivity in a developmentally dependent manner that builds on previous studies. In particular, several previous studies reported animal and human evidence on the role of IPS in visuomotor functions (*Grefkes et al., 2004*; *Snyder et al., 1997*; *Andersen et al., 1998*; *Battaglia-Mayer and Caminiti, 2002*). Importantly, the human IPS was shown to subserve visuomotor transformation independently of the modality-specific processing of visual or proprioceptive information (*Grefkes et al., 2004*). Here, we propose a generic, resting-state mechanistic model by which the levels of excitation and inhibition within the left IPS shape task-independent visuomotor functions through the visuomotor resting-state connectivity across

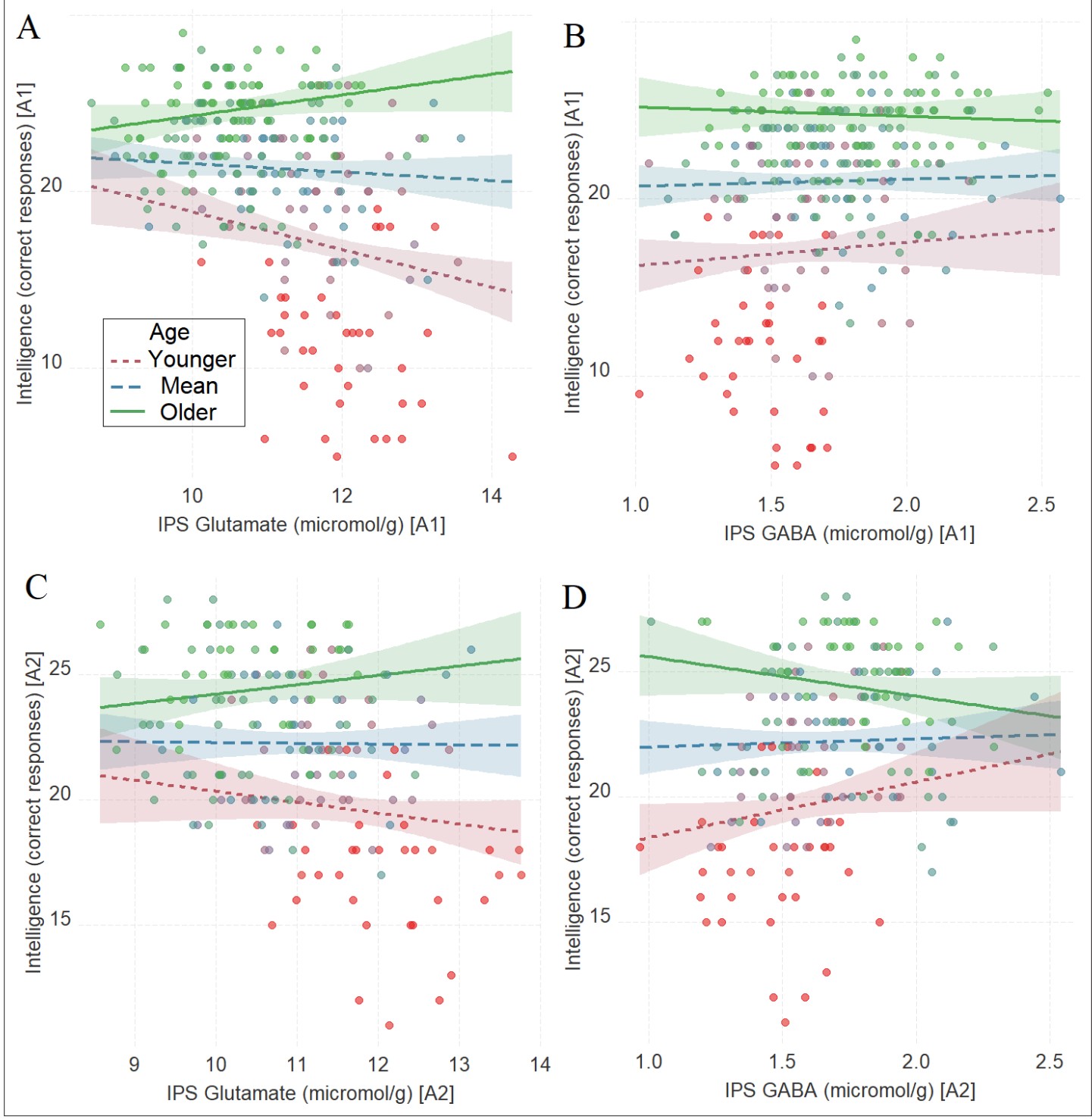

**Figure 6.** The moderating role of age in the relation between neurotransmitter concentration and intelligence at A1 (**A, B**) and A2 (**C, D**) in Task 4. (**A**) Intraparietal sulcus (IPS) glutamate * age and (**B**) IPS GABA * age at A1; (**C**) IPS glutamate * age and (**D**) IPS GABA * age at A2. We utilized a standard practice for examining interactions with continuous variables using three levels of a moderator (*Aiken et al., 1991*). Specifically, to depict the interaction between the continuous variables (age and neurotransmitter concentration), we plotted the regression lines for younger and older participants (i.e., ±1 SD) from the mean age (*Aiken et al., 1991*). Green colors represent +1 SD above the mean (older participants), red colors represent −1 SD below the mean (younger participants), and blue colors represent the mean. Since our cohort was based on a developmental sample spanning from early childhood to early adulthood, 'younger participants' refer predominantly to early and late childhood and 'older participants' refer predominantly to late adolescence and early adulthood. The shaded area represents 95% confidence intervals. In these multiple regression analyses, the sample sizes were 260, 182, 261, and 182, for panels A, B, C and D respectively, and the significance level was set to 0.05 (5%).

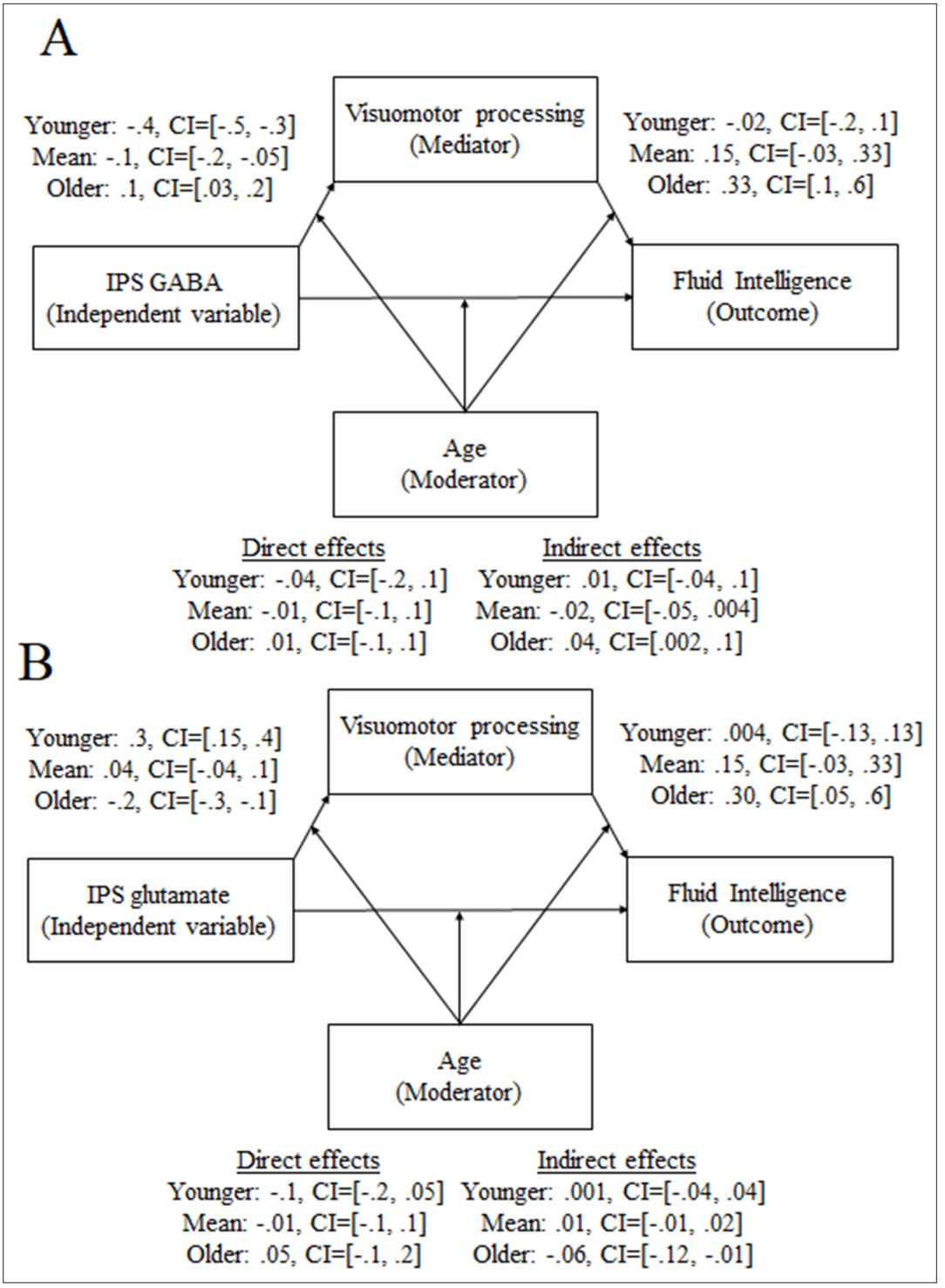

**Figure 7.** Moderated mediation results on the role of visuomotor processing in the relation between fluid intelligence and intraparietal sulcus (IPS) GABA (**A**) or IPS glutamate (**B**) only for the older participants.

development. However, several aspects of this proposed model are still not known. For example, are there specific time windows during visuomotor processing when the levels of IPS excitation and inhibition regulate the activity of visuomotor regions? Do the levels of IPS excitation and inhibition regulate the activity of visuomotor regions by regulating local GABA and glutamate within visuomotor regions? Addressing these questions is beyond the scope of this study but can be examined in an investigation combining functional MRS and task-based fMRI (*Jia et al., 2022*; *Koolschijn et al., 2021*).

## Visuomotor processing mediates the relationship between IPS neurochemicals and fluid intelligence depending on the developmental stage

In a previous study, age differences in intelligence were accounted for by age differences in visuo-motor processing (*von Krause et al., 2020*). This raises the intriguing possibility that the developmental link we found between the IPS glutamate and GABA and fluid intelligence may be mediated by visuomotor processing. Indeed, we found that individual variation in visuomotor processing mediated the relationship between IPS glutamate and GABA levels and fluid intelligence in a developmentally dependent manner. Specifically, visuomotor processing mediated the relationship between IPS glutamate and GABA and fluid intelligence only for the older participants. Therefore, our study provides the neurobiological root that may drive this association: IPS glutamate and GABA. It is currently an open question why this relationship was observed only for the older participants. One potential explanation is that visuomotor processing becomes relevant and mediates the relationship between IPS neurochemicals and fluid intelligence only when the individual's visuomotor processing has matured to the point that it becomes relatively automated and stabilized, which is the case with the older participants in our sample. Regarding the underlying neurobiological mechanisms, visuo-motor processing becomes relevant in tracking fluid intelligence only when the visuomotor functions of the IPS have reached relative maturity. Specifically, the GABA- and glutamate-dependent maturity of IPS neurons that respond to the visuomotor domain may determine the onset of the influence of visuomotor processing on fluid intelligence (*Grefkes and Fink, 2005*). An alternative suggestion is that the strategies used to solve fluid intelligence tasks (such as the one used in the current study) differ as a function of development and are subserved by different mechanisms.

## Quantitative and qualitative differences in how participants solve fluid intelligence problems

It is noteworthy that even though the other two diffusion parameters (mean drift rate and boundary separation) are not related to IPS neurochemicals, the relationship between all three diffusion parameters and fluid intelligence was moderated by age(see *Figure 2—figure supplement 2* and *Supplementary file 8*). In particular, mean drift rate explained fluid intelligence only for the younger and the mean age groups, boundary separation explained fluid intelligence only for the younger age group, and visuomotor processing explained fluid intelligence only for the older age group. Moreover, the IPS is involved in fluid intelligence and visuomotor processing in later developmental stages (i.e., for the older but not the younger participants in our sample). Please note that even though lower signal quality may explain the failure to detect these relationships for certain groups of participants (e.g., younger participants in this instance), as can be seen in 'Materials and methods,' this was not the case as the average signal quality was higher in younger participants (where this relationship was not detected) compared to older participants (where this relationship was detected).

Taken together, our results show that some of the mechanisms in fluid intelligence are based on cognitive (mean drift rate), decision (boundary separation), and visuomotor processing (non-decision time), and the involvement of each of these mechanisms depends on age. These findings suggest qualitative and quantitative differences in the way that participants across development solve fluid intelligence problems of the kind presented in our research.

A limitation of this study, also common to other MRS studies, relates to the difficulty in distinguishing between intracellular and extracellular neurotransmitter concentrations or even a portion of these based on the MRS signal alone (*Dyke et al., 2017*). Consequently, making direct inferences of cortical excitability/inhibition based on the neurotransmitter concentrations alone should be done cautiously. Similar limitations concerning the exact nature of the neural signal are not specific to MRS but exist with other modalities, for example, diffusion and structural MRI (*Zatorre et al., 2012*; *Logothetis et al., 2001*).

Another limitation stems from the fact that using an unedited sequence is not ideal if one aims to merely quantify GABA. We chose an unedited approach since we wanted to acquire metabolite data (not only the GABA) with the shortest possible acquisition duration to minimize possible motion artifacts during acquisition. The unedited method provided high-quality spectra from an 8 mL VOI in a procedure that takes 10–15 min. In addition, the unedited method minimizes the vulnerability of the acquisition method to system imperfections (i.e., magnetic field drift), subject motions, and

magnetic field inhomogeneity. Alternatively, one can choose to use an edited method with a larger VOI (27 mL) and a longer acquisition duration. Because our study was focused on quantifying both GABA and glutamate, we utilized a sequence capable of detecting both these neurotransmitters. For example, previous studies have shown the method we employed can detect GABA in a valid and reliable manner (*Frangou et al., 2019*; *Ip et al., 2019*; *Hong et al., 2019*; *Joers et al., 2018*; *Kolasinski et al., 2017*; *Barron et al., 2016*), and a study demonstrated that short-TE MRS can be employed for the reproducible detection of GABA at 3T (*Near et al., 2013*). Moreover, as was discussed in the article, we excluded cases where the Cramér–Rao lower bound (CRLB) was poor based on previously established criteria, thus ensuring that our quantification of single metabolites (e.g., GABA) was appropriate.

Another related issue of low spectral resolution is the challenge of reliably separating certain metabolites which exhibit substantial spectral overlap with each other and with other strong signals as well as the underlying macromolecular background. Of note, the mean cross-correlation between the neurotransmitter pairs glutamate-Gln and glutamate-GABA in both regions of interest (MFG and IPS) was <0.5, suggesting that there was no significant overlap between these neurotransmitters and allows to report these concentrations separately (*Zacharopoulos et al., 2021c*; *Zacharopoulos et al., 2021a*). Furthermore, the cross-correlation between GABA concentration and the concentration of any other metabolite in both regions was not particularly high (i.e., <0.5). Taken this empirical evidence together, our sequence was able to detect and quantify GABA and glutamate appropriately.

Another limitation of this study regarding the computation of the mean drift rate, boundary separation, and visuomotor processing from the relatively low number of trials (although a larger number of participants) and the high accuracy rate (*Supplementary file 9*) of the tasks used, both of which can negatively impact the reliability of parameter estimates. However, it was previously demonstrated that the EZ-diffusion model's ability to accurately reflect individual differences in model parameters (both in large, 800 trials, and in small, 80 trials, data sets) and its ability to recover the experimental effects on parameter means (*van Ravenzwaaij and Oberauer, 2009*) often outperformed that of other diffusion models such as fast-dm (*Voss and Voss, 2007*) and DMAT (*Vandekerckhove and Tuerlinckx, 2007*).

Visuomotor processing, which previously showed limited evidence of psychometric validity as a task-general trait (*Schubert et al., 2016*), is (theoretically) the summation of a wide array of different processes such as perceptual encoding, visual search, and motor responding, and this study does not dissociate between these. Therefore, it is difficult to assert that visuomotor processing assessed by this study alone indexes a unitary mechanistic process. Rather, future work that further dissociates the components of visuomotor processing can shed some light on the underlying visuomotor processes that drive the effects observed here.

In sum, we have shown that glutamate and GABA within the IPS track visuomotor functions across development. Using additional analyses, we have established that visuomotor resting-state connectivity mediates the relationship between IPS neurochemicals and visuomotor processing, and that visuomotor processing mediates the relationship between IPS glutamate and GABA and fluid intelligence. These findings provide a mechanistic understanding of the involvement of GABA and glutamate in the IPS, and their potential contribution to other cognitive functions, such as mathematical development (*Zacharopoulos et al., 2021c*), which has been suggested to be subserved partly by visuomotor functions (*Cohen Kadosh et al., 2008*; *Walsh, 2003*). In addition, our study provides support for the view that fluid intelligence is also subserved by visuomotor processes, in this case, in the older participants in our sample, and the involvement of glutamate and GABA. It is known that the brain develops and specializes to support such high-level cognitive functions (*Anderson, 2008*; *Dehaene and Cohen, 2007*; *Johnson, 2001*). An integrative and robust understanding of psychology and biology across the different developmental stages, as we show in this study, would allow the researcher to progress toward personalized learning and precise diagnosis and intervention (*Dockterman, 2018*; *Schork, 2015*).

## Materials and methods

### Participants

We recruited 293 participants, and the demographic information for both the first assessment and the second assessment is reported in *Supplementary file 1*. The imaging session each lasted ~60 min, and the cognitive and behavioral tasks described here each lasted ~30 min and were part of a larger battery of tests. All imaging data were acquired in a single scanning session during which participants watched the LEGO Movie (*Phill, 2014*), except for the data acquired in the resting fMRI session (see below) during which participants were asked to fixate on a white cross displayed in the center of a black background. All participants were predominantly right-handed, as measured by the Edinburgh Handedness Inventory (*Oldfield, 1971*), and self-reported no current or past neurological, psychiatric, or learning disability or any other condition that might affect cognitive or brain functioning. For every assessment (i.e., first, second), adult participants received £50 compensation for their time, and child participants, depending on their age, received £25 (early childhood) and £35 (late childhood, early adolescence, late adolescence) in Amazon or iTunes vouchers, and additional compensation for their caregiver if the participant was under the age of 16. Informed written consent was obtained from the primary caregiver and informed written assent was obtained from participants under the age of 16, according to approved institutional guidelines. The study was approved by the University of Oxford's Medical Sciences Interdivisional Research Ethics Committee (MS-IDREC-C2_2015_016).

### Magnetic resonance imaging

All MRI data were acquired using a 3T Siemens MAGNETOM Prisma MRI System equipped with a 32-channel receive-only head coil. Anatomical high-resolution T1-weighted scans were acquired consisting of 192 slices, repetition time (TR) = 1900 ms; echo time (TE) = 3.97 ms; voxel size = 1 × 1 × 1 mm. Simulations were performed using the same RF pulses and sequence timings as in the 3T system described above (*Terpstra et al., 2016*). The MRspa preprocessing steps included the following: (1) cross-correlation method to minimize the frequency difference between single-shot MRS data, (2) least-square method to minimize the phase difference between single-shot MRS data, (3) eddy current correction (ECC) was used to correct for both ECC based on the phase of water reference scan, and (4) save the averaged spectrum.

### Magnetic resonance spectroscopy

Spectra were measured by semi-adiabatic localization using an adiabatic selective refocusing (semi-LASER) sequence (TE = 32 ms; TR = 3.5 s; 32 averages) (*Deelchand et al., 2015*; *Oz and Tkáč, 2011*) and variable power RF pulses with optimized relaxation delays (VAPOR), water suppression, and outer volume saturation. Unsuppressed water spectra acquired from the same volume of interest were used to remove residual eddy current effects and reconstruct the phased array spectra with MRspa (https://www.cmrr.umn.edu/downloads/mrspa/). Two 20 × 20 × 20 mm$^3$ voxels of interest were manually centered in the left IPS and the MFG based on the individual's T1-weighted image while the participant was lying down in the MR scanner. Acquisition time per voxel was 10–15 min, including manual voxel placement, sequence planning, shimming, potential readjustments if necessary, and the acquisition of the water T2 series (see below).

MRS neurotransmitters were quantified with LCModel (*Provencher, 2001*) using a basis set of simulated spectra generated based on previously reported chemical shifts and of coupling constants generated based on a VeSPA (versatile simulation, pulses, and analysis) simulation library (*Soher et al., 2011*). Simulations were performed using the same RF pulses and sequence timings as in the 3T system described above. Overall, we utilized one of the most established methods in the literature since the early 2000s, including the LCModel analysis (*Tkáč et al., 2001*). Moreover, as shown previously, our methods produced robust test–retest reliability in a previous study (*Terpstra et al., 2016*) where glutamate NAA, tCr, tCho, and Ins were quantified with between-session coefficients of variance of ≤5% even at 3T, and GSH mimicking the same GABA condition had a coefficient of variance of ~10%. Raw neurochemical concentrations (i.e., metabolite-to-water amplitude ratios) were extracted from the spectra using a water signal as an internal concentration reference. We used the LCModel default output with the following parameters: ATTMET = 32, ATTH2O = 1, WCONC = 55555, PPMSHF = 0.0, PPMEND = 0.5, PPMST = 4.2, DKNTMN = 0.25, RFWHM = 2.5, SDDEGP = 5.00, SDDEGZ = 5.00, DOWS = TRUE, DOECC = FALSE, DELTAT = 0.0001666, NUNFIL = 2048,

HZPPPM = 123.2571, DEGPPM = 0, DEGZER = 0. For a stacked plot depicting the spectrum, the fit, the residual, the baseline estimation, and the individual contributions from all basis functions, see *Figure 1—figure supplement 1*.

The exclusion criteria for data were (i) CRLB and (ii) the signal-to-noise balance (SNR). Neurochemicals quantified with CRLB (the estimated error of the neurochemical quantification) >50% were classified as not detectable. We aimed to follow a relatively unbiased approach, avoid using hard thresholds issues, and adopt the suggested procedure highlighted previously (*Kreis, 2016*). In addition, since GABA concentration is relatively low (compared to glutamate), which usually induced high CRLB values, and since this study was mainly focused on GABA and glutamate, a good compromise was to exclude CRLB > 50%. Additionally, we excluded cases with an SNR beyond 3 SDs per region. By performing bivariate correlations, we found that older (vs. younger) participants exhibited lower SNR (p<0.001), higher line width (p<0.001), and higher glutamate CRLB (p<0.001) across both regions, which is expected as we have shown previously that older participants exhibit lower T2 values (*Zacharopoulos et al., 2021a*). We also excluded cases with a neurochemical, connectivity (see below), or behavioral score beyond 3 SDs (per age group), and cases where the standardized residuals in a given analysis were beyond 3 SDs.

Raw neurochemical concentrations were then scaled based on the structural properties of the selected regions and based on the predefined values shown in *Equations 1 and 2* (see below) (*Provencher, 2001*); these predefined correction values were therefore determined before the data collection. By predefined correction values, we specifically refer to the values 43,300, 35,880, and 55,556, which were the water concentrations in mmol/L for gray matter (GM), white matter (WM), and cerebrospinal fluid (CSF) (*Equation 1*). On the other hand, the GM, WM, and CSF fractions (*Equation 1*) and T2 of tissue water value (*Equation 2*) were not determined prior to data collection. Instead, these values were calculated in a subject-specific manner as described below. To quantify the structural properties, we segmented the images into different tissue classes including GM, WM, and CSF using the SPM12 segmentation facility. Next, we calculated the number of GM, WM, and CSF voxels within the two masks of interest separately around the left IPS and MFG in native space. Subsequently, we divided these six numbers (GM, WM, and CSF for IPS and MFG, respectively) by the total number of GM, WM, and CSF voxels to obtain the corresponding GM, WM, and CSF fraction values per participant and region. As a final computation step, we scaled the raw neurotransmitter values to these structural fractions using the following LCModel (*Provencher, 2001*) computation:

$$\text{Tissue corrected concentration} = ((43300/55556^*\text{GM fraction} + 35880/55556\ ^*\text{WM fraction} + 1\ ^*\text{CSF fraction})/(1 - \text{CSF fraction}))\ ^*\text{raw neurochemical concentration} \tag{1}$$

This was done to perform partial-volume correction based on three tissue classes as mentioned above. The numerator is the total water concentration in the region of interest, and the factor 1/(1-CSF fraction) is the partial volume correction in that it corrected for the fact that all neurochemicals are concentrated in the GM and WM. The values 43,300, 35,880, and 55,556 are the water concentrations in mmol/L for GM, WM, and CSF, respectively, and these were the default water concentration values employed by the LCModel (*Provencher, 2014*). The numerator corrected for differing tissue water concentrations for the unsuppressed water reference, whereas the denominator corrected for the assumption that CSF is free of metabolites.

In the main text we present the results of the tissue corrected concentration values (*Equation 1*). To minimize the potential confounding effects T2-relaxation times, we additionally report the results (*Supplementary file 13*) where we corrected the concentration from *Equation 1* based on the T2 of tissue water values. These concentration values were scaled based on the T2 of tissue water values as can be seen in *Equation 2* below. Fully relaxed unsuppressed water signals were acquired at TEs ranging from 32 to 4040 ms (TR = 15 s) to water T2 values in each region of interest (32 ms, 42 ms, 52 ms, 85 ms, 100 ms, 115 ms, 150 ms, 250 ms, 450 ms, 850 ms, 1650 ms, 3250 ms, 4040 ms). The transverse relaxation times (T2) of tissue water and the percent CSF contribution to the region of interest were obtained by fitting the integrals of the unsuppressed water spectra acquired in each region of interest at different TE values with a biexponential fit (*Piechnik et al., 2009*), with the T2 of CSF fixed at 740 ms and three free parameters: T2 of tissue water, amplitude of tissue water, and amplitude of CSF water.

$$\text{T2 corrected concentration} = \text{tissue corrected concentration} * \exp(-\text{TE/T2}) \tag{2}$$

The results showed the same general pattern across all quantification methods as can be seen in *Supplementary files 11–13*. The number of spectral data points for the MFG and the IPS was 2048, and the spectral bandwidth was 6000 Hz.

## Resting fMRI

Functional images were acquired with a multi-band acquisition sequence (multi-band accel. factor = 6, TR = 933 ms, TE = 33.40 ms, flip angle 64°, number of slices = 72, voxel dimension = 2 × 2 × 2 mm, number of volumes = 380). fMRI data were preprocessed and analyzed using the CONN toolbox ( www.nitrc.org/projects/conn, RRID:SCR_009550) in SPM12 (Wellcome Department of Imaging Neuroscience, Institute of Neurology, London, UK) using the default preprocessing pipeline 'MNI-space direct normalization' (*Whitfield-Gabrieli and Nieto-Castanon, 2012*). Functional volumes were motion-corrected, slice-time corrected, segmented, normalized to a standardized (MNI) template, spatially smoothed with a Gaussian kernel (8 mm FWHM), and pass filtered (0.01 Hz to Inf). The low-pass portion of the filter (0.01 Hz) was used to reduce the physiological and noise components contributing to the low-frequency segment of the BOLD signal, and the high-pass portion of the filter was set to Inf to additionally allow higher frequencies to be included (above 0.1 Hz) given our fast acquisition. This also increased the degrees of freedom, thereby improving the quality of denoising and connectivity estimation. We also accounted for physiological noise in the time series by regressing out the confounding effects of WM, CSF, realignment, and scrubbing that were automatically calculated in CONN. We calculated the functional connectivity of the visuomotor network (which is featured in CONN; *Schubert et al., 2016*), which consisted of seven regions, four of which were derived from the visual network (networks.Visual.Medial (2, −79, 12), networks.Visual.Occipital (0, −93, −4), networks.Visual.Lateral [L] (−37, −79, 10), networks.Visual.Lateral [R] (38, −72, 13)), the three of which were derived from the motor network (networks.SensoriMotor.Lateral [L] (−55, −12, 29), networks.SensoriMotor.Lateral [R] (56, −10, 29), networks.SensoriMotor.Superior (0, −31, 67)), yielding a single visuomotor functional connectivity score for each participant, which we refer to as visuomotor connectivity. The motion scrubbing threshold was 2 mm. We excluded cases where more than 5% of the scans were removed, and the number of cases removed for each age group was as follows: early childhood = 29, late childhood = 4, early adolescence = 1, later adolescence = 3, and early adulthood = 0. After selecting the regions making up the visuomotor network, we used the CONN toolbox to calculate the within-network connectivity (averaged across all the ROIs within the visuomotor network) for each participant. The connectivity values presented are Fisher r-to-z-transformation.

## Attention network task (Task 1)

Participants completed the child version of the original attention network task (*Fan et al., 2002*), and the stimuli were presented using E-Prime on a desktop computer (*Figure 1B*). We included a single block of trials in the analyses (96 trials, 12 conditions) as this was the block that all age groups completed. We excluded trials with (i) a reaction time of <200 ms and/or with (ii) a reaction time that was beyond 3 SDs from the individual mean reaction time across all conditions. We calculated three behavioral measures (three diffusion parameters: mean drift rate, boundary separation, and visuomotor processing). The three diffusion parameters were calculated following the procedure outlined in the original work (*Wagenmakers et al., 2007*) by averaging the mean reaction time of all correct trials without focusing on predefined conditions (i.e., they were estimated from trial-level data that were collapsed across conditions). For the results of the three attention network tasks separately, see the supplementary experiment below. The motivation of the supplementary experiment was to provide the reader with complementary information of not only how the generic diffusion parameters (i.e., quantified across all trials without taking into account the condition) reported in the main text are related to neurochemicals over development but rather how the condition-based diffusion parameters (e.g., in response to a particular attention network) are related to neurobiology and development.

## Digit comparison task (Task 2)

In the digit comparison task, participants were asked to choose the larger of two single-digit Arabic numbers as quickly and accurately as possible by clicking the mouse button corresponding to the side of the selected number (*Figure 1B*). The numbers remained on the screen until the participant

responded. Between each trial, a central fixation hashtag appeared for 1000 ms. There were two practice trials followed by 72 test trials entailing all 36 possible comparisons of numbers between 1 and 9 repeated twice. The larger number appeared on the left side of the screen in half of the trials. For the results of the distance effect, see the supplementary experiment in 'Materials and methods' section.

### Mental rotation task (Task 3)

In this task, participants decided whether a rotated target figure presented on the right side of the screen was the same (i.e., non-mirrored) or different (i.e., mirrored) compared to the upright figure presented on the left side of the screen (*Figure 1B*). Each trial began with a black fixation cross in the center of the white screen for 1000 ms. Thereafter, the two figures (upright and rotated) appeared on the screen, and participants responded by pressing the left or right mouse button, where the mouse button presses associated with the labels 'same' and 'different' were counterbalanced. The labels 'same' and 'different' appeared at the bottom left and bottom right of the screen, respectively, to remind participants which mouse button to press if the figures were the 'same' or 'different.' Participants were instructed to avoid mistakes and respond as fast as they could. The figures remained on the screen until response and then were followed by a scrambled mask for 150 ms. There were two sets composed of three figures each (animals set: giraffe, zebra, raccoon; letters set: R, F, G). There were 24 trials for each figure, in which the rotated figure changed its pointing direction (left and right), degree of rotation (45, 90, 135), direction of rotation (clockwise and anticlockwise), and the rotated figure was either the same or different (non-mirrored or mirrored) compared to the upright figure. At the beginning of each block, participants completed 10 practice trials, and they were allowed to take a break in the middle of each block. We focused on the 'animal trials' as only these trials were present in all age groups. For the results of the distance effect, see the supplementary experiment in 'Materials and methods' section. For the associations between the diffusion parameters within and between the three tasks, see *Supplementary file 10*.

### Fluid intelligence task (Task 4)

Participants completed the matrix reasoning subtest of the Wechsler Abbreviated Scale of Intelligence (WASI II) (*Wechsler, 2011*) as an index of fluid intelligence (*Figure 1B*). The subset of matrix reasoning is a 30-item (of increasing difficulty) assessment tool that requires identifying a logical pattern in a sequence of visuospatial stimuli. The subtest is interrupted when the participant provides three consecutive wrong responses. We calculated the number of correct answers.

### Statistical analyses

Chronological age, which was treated as a continuous variable in the statistical analyses, spanned from early childhood (6–7 y), late childhood (10–11 y), early adolescence (14–15 y), late adolescence (16–17 y), to early adulthood (18+ y). For additional information on the mean age, see *Supplementary file 1*. Since chronological age was the main moderator in this study, we termed +1 SD above the mean as 'older participants,' –1 SD below the mean as 'younger participants,' and the mean as the 'mean.' Since our cohort was based on a developmental sample spanning from early childhood to early adulthood, 'younger participants' refer predominantly to early and late childhood and 'older participants' refer predominantly to late adolescence and early adulthood. The dependent variables were three behavioral measures (i.e., the three diffusion parameters: mean drift rate, boundary separation, and visuomotor processing). For the primary neurochemical analyses, the independent variables of interest were (i) the neurochemical concentration and (ii) the interaction between the neurochemical concentration and age. The former independent variable assesses whether there is an overall effect of neurochemical concentration on each diffusion parameter, while the latter independent variable considers the moderating role of age in shaping the relation between neurotransmitter concentration and that diffusion parameter. In the text we present the standardized coefficients. These were obtained by z-scoring the continuous predictors and continuous dependent variables. Importantly, in every model with a diffusion parameter (e.g., mean drift rate) as the dependent variable, we controlled for the other two diffusion parameters (e.g., boundary separation and visuomotor processing), as can be seen in *Equations 3–5*, as this helped to assess whether the results for one diffusion parameter were not confounded by the other two diffusion parameters. To assess these effects during the first assessment (A1), we employed *Equation 3* (see below); to assess the same associations during the

second assessment (A2), we employed *Equation 4*; and to assess the same associations during the prediction/longitudinal analyses, we employed *Equation 5*. The prediction/longitudinal analyses were conducted to assess whether neurochemical concentration and its confluence with age during the first assessment can predict three behavioral measures (i.e., mean drift rate, boundary separation, and visuomotor processing) obtained during the second assessment. In the following equations, the 'behavioral score' is always one of the three diffusion parameters, and the control behavioral scores 1 and 2 are the other two diffusion parameters.

$$
\begin{aligned}
behavioral\ score\ (A1) \quad \sim \quad & neurotransmitter\ (A1) \ + age\ (A1) \ + neurotransmitter\ (A1) * age\ (A1) \ + \\
& control\ behavioral\ score\ 1\ (A1) \ + control\ behavioral\ score\ 2\ (A1)
\end{aligned}
\tag{3}
$$

$$
\begin{aligned}
behavioral\ score\ (A2) \quad \sim \quad & neurotransmitter\ (A2) \ + age\ (A2) \ + neurotransmitter\ (A2) * age\ (A2) \ + \\
& control\ behavioral\ score\ 1\ (A2) \ + control\ behavioral\ score\ 2\ (A2)
\end{aligned}
\tag{4}
$$

$$
\begin{aligned}
behavioral\ score\ (A2) \quad \sim \quad & neurotransmitter\ (A1) \ + age\ (A1) \ + neurotransmitter\ (A1) * age\ (A1) \ + \\
& control\ behavioral\ score\ 1\ (A1) \ + control\ behavioral\ score\ 2\ (A1) \ + control\ behavioral\ score\ 1\ (A2) \ + \\
& control\ behavioral\ score\ 2\ (A2) \ + age\ (A2)
\end{aligned}
\tag{5}
$$

To correct for multiple comparisons during the first assessment, we performed FDR correction correcting for 36 comparisons [diffusion parameters (3: drift rate, boundary separation, visuomotor processing) * neurochemical (3: GABA, glutamate, NAA) * region (2: IPS, MFG) * type of effect (2: neurochemical * age interaction, the main effect of neurochemical)]. To correct for multiple comparisons during the prediction/longitudinal analyses, we performed FDR correction correcting for 36 comparisons [diffusion parameters (3: drift rate, boundary separation, visuomotor processing) * neurochemical (3: GABA, glutamate, NAA) * region (2: IPS, MFG) * type of effect (2: neurochemical * age interaction, the main effect of neurochemical)]. From the longitudinal analyses (Task 1), the only association that survived the FDR correction was the IPS GABA statistical model predicting visuomotor processing, but this did not survive neurochemical and neuroanatomical specificity.

Since we applied a separate FDR multiple comparison correction for the first assessment analyses in Task 1 and for the prediction/longitudinal analyses in Task 1, we did not correct for multiple comparisons for all replication analyses, nor did we assess the (neurochemical, neuroanatomical and diffusion-parameter) specificity in the replication analyses of Task 2 and Task 3 to reduce type 2 errors. Regarding the supplementary experiment (for details, see 'Materials and methods' section), we applied a separate FDR multiple correction correcting for 180 comparisons during the first assessment and a further separate FDR multiple correction correcting for 180 comparisons during the longitudinal/prediction analyses [diffusion parameters (3: mean drift rate, boundary separation, visuomotor processing) * neurochemical (3: GABA, glutamate, NAA) * region (2: IPS, MFG) * type of effect (2: neurochemical * age interaction, the main effect of neurochemical) * cognitive function (5: alerting network, orienting network, executive network, distance effect, SNARC effect)].

Apart from correcting for multiple comparisons, we assessed the neurochemical and neuroanatomical specificity of our findings that survived the aforementioned FDR correction. To this end, we ran multiple regression models with several independent variables, namely the main effect of all six neurotransmitter measures (i.e., IPS glutamate, IPS GABA, IPS NAA, MFG glutamate, MFG GABA, MFG NAA) and their interactions with age, as well as the main effect of age as can be seen in *Equation 6*(A1), *Equation 7* (A2), and *Equation 8* (prediction/longitudinal) below.

$$
\begin{aligned}
behavioral\ score\ (A1) \ \sim \ & IPS\ glutamate\ (A1) * age\ (A1) \ + IPS\ GABA\ (A1) * age\ (A1) \ + IPS\ NAA\ (A1) * \\
& age\ (A1) \ + MFG\ glutamate\ (A1) * age\ (A1) \ + MFG\ GABA\ (A1) * age\ (A1) \ + MFG\ NAA\ (A1) * age\ (A1) \ + \\
& IPS\ glutamate\ (A1) \ + IPS\ GABA\ (A1) \ + IPS\ NAA\ (A1) \ + MFG\ glutamate\ (A1) \ + MFG\ GABA\ (A1) \ + \\
& MFG\ NAA\ (A1) \ + age\ (A1)
\end{aligned}
\tag{6}
$$

$$\begin{aligned}
behavioral\ score\ (A2) \sim\ &IPS\ glutamate\ (A2) * age\ (A2)\ +\ IPS\ GABA\ (A2) * age\ (A2)\ +\ IPS\ NAA\ (A2) * \\
&age\ (A2)\ +\ MFG\ glutamate\ (A2) * age\ (A2)\ +\ MFG\ GABA\ (A2) * age\ (A2)\ +\ MFG\ NAA\ (A2) * age\ (A2)\ + \\
&IPS\ glutamate\ (A2)\ +\ IPS\ GABA\ (A2)\ +\ IPS\ NAA\ (A2)\ +\ MFG\ glutamate\ (A2)\ +\ MFG\ GABA\ (A2)\ + \\
&MFG\ NAA\ (A2)\ +\ age\ (A2)
\end{aligned} \tag{7}$$

$$\begin{aligned}
behavioral\ score\ (A2) \sim\ &IPS\ glutamate\ (A1) * age\ (A1)\ +\ IPS\ GABA\ (A1) * age\ (A1)\ +\ IPS\ NAA\ (A1) * \\
&age\ (A1)\ +\ MFG\ glutamate\ (A1) * age\ (A1)\ +\ MFG\ GABA\ (A1) * age(A1)\ +\ MFG\ NAA\ (A1) * age(A1)\ + \\
&IPSglutamate\ (A1)\ +\ IPS\ GABA\ (A1)\ +\ IPS\ NAA\ (A1)\ +\ MFG\ glutamate\ (A1)\ +\ MFG\ GABA\ (A1)\ + \\
&MFG\ NAA\ (A1)\ +\ age\ (A1)\ +\ age\ (A2)
\end{aligned} \tag{8}$$

For the results from **Equations 3 and 4** see **Supplementary file 3**.

In Tasks 1–3, the visuomotor processing score was calculated based on each task accordingly. Namely, the visuomotor processing in Task 1 was computed based on the attention network task (Task 1), the visuomotor processing in Task 2 was computed based on the digit comparison task (Task 2), and the visuomotor processing in Task 3 was computed based on the mental rotation task (Task 3). Since we obtained the same pattern of findings regarding visuomotor processing across Tasks 1–3, which was one of our earlier aims, for the visuomotor connectivity analyses and all moderated mediation analyses, we computed a composite visuomotor processing score that was obtained based on Tasks 1–3. Specifically, we z-scored the visuomotor processing in each of the three tasks separately and calculated the mean of these three scores which represents a generic or task-independent visuomotor processing score for each participant (of note, we ran related and complementary factor analyses reported in **Supplementary file 14**).

To assess whether the visuomotor connectivity interacted with age in explaining visuomotor processing, we ran the multiple regression model displayed below in **Equation 9** .

$$\begin{aligned}
Visuomotor\ processing\ (A1) \sim\ &visuomotor\ connectivity\ (A1)\ +\ age\ (A1)\ +\ visuomotor\ connectivity\ (A1) * \\
&age\ (A1)
\end{aligned} \tag{9}$$

To assess whether the neurotransmitter scores interacted with age in explaining the visuomotor connectivity, we ran a multiple regression model displayed below in **Equation 10**.

$$Visuomotor\ connectivity\ (A1) \sim\ neurotransmitter\ (A1)\ +\ age\ (A1)\ +\ neurotransmitter\ (A1) * age\ (A1) \tag{10}$$

The statistical results presented in the 'Results' section involve multiple linear regressions with bootstrapping (5000 samples) and show the uncorrected p-value denoted by $P_{BO}$, as well as the lower (CI_L) and upper (CI_U) bounds of the confidence intervals (CIs) obtained from bootstrapping.

To examine the possibility that IPS neurochemicals (i.e., glutamate and/or GABA) are associated with visuomotor processing via visuomotor connectivity across development, we ran a moderated mediation model (model 59, PROCESS v4.1, **Hayes, 2017**) where we evaluated whether the neurotransmitter concentration (independent variable) tracks visuomotor processing (dependent variable) via the visuomotor connectivity (mediator) as a function of development (moderator).

In these mediation analyses and all other multiple regression models, we additionally removed cases that fell beyond 3 SDs after running the multiple regression models (in this case, **Equation 11** ).

$$\begin{aligned}
Visuomotor\ processing\ (A1) \sim\ &neurotransmitter\ (A1)\ +\ age\ (A1)\ +\ visuomotor\ connectivity\ (A1)\ + \\
&neurotransmitter\ (A1) * age\ (A1)\ +\ visuomotor\ connectivity\ (A1) * age\ (A1)
\end{aligned}$$

$$\tag{11}$$

To examine the possibility that IPS neurochemicals were associated with intelligence via visuomotor processing across development, we ran a moderated mediation model (model 59, PROCESS v4.1, **Hayes, 2017**) where we evaluated whether the neurotransmitter concentration (independent variable) tracks intelligence (dependent variable) via the visuomotor processing (mediator) as a function of development (moderator) (**Equation 12**). We assessed the statistical significance by examining the indirect effect at 90% CIs (and not all individual paths) with bootstrapping (5000 samples). We acknowledge that the mediation analyses were not part of the original aims of the study (i.e., the interrelations of diffusion parameters and neurochemicals over development assessed in Task 1) and were

made post hoc after the original hypotheses were confirmed. Relatedly, we used 90% CI to evaluate the significance of effects in the mediation analyses, and the p-values were uncorrected.

$$Fluid\ intelligence\ (A1)\ \sim\ neurotransmitter\ (A1)\ +\ age\ (A1)\ +\ visuomotor\ processing\ (A1)\ +$$
$$neurotransmitter\ (A1) * age\ (A1)\ + visuomotor\ processing\ (A1)\ * age\ (A1)$$

(12)

## Supplementary experiment

Attention is vital for daily functioning and plays a critical role in human development across multiple domains, such as perception, language, and memory (*Posner et al., 2016*; *Rosario Rueda et al., 2015*). Altered attention processing manifests in severe disorders including, but not limited to, ADHD, schizophrenia, borderline personality disorder, and brain injury (e.g., stroke) (*Fan and Posner, 2004*).

Several models have presented attention as a multifaceted construct typically divided into several functional systems (*Petersen and Posner, 2012*). In particular, Posner's model of attention entails three systems: (i) alerting, (ii) orienting, and (iii) executive (*Fan et al., 2002*; *Fan et al., 2003*). The altering system is divided into two subcomponents: tonic and phasic (*Rueda et al., 2015*). Tonic alerting corresponds to sustained attention or vigilance, which is needed to respond to low-frequency events rapidly (*Ratcliff et al., 2016*). On the other hand, phasic alertness corresponds to the advantage in processing stimuli preceded by warning cues compared with those that are not (*Fan et al., 2002*). The orienting system represents individuals' ability to move attention to a spatial location (*Rosario Rueda et al., 2015*). The executive system allows allocating attention to objects or events despite conflicting information and allows switching among tasks.

All three attention systems exist to some degree in infancy and change over life span (*Rueda et al., 2004*). Cross-sectional studies have suggested that the executive system stabilizes around the age of 7, alertness improves even beyond the age of 10, whereas orienting remains stable (*Rueda et al., 2004*). Other studies, examining children from 6 to 12 y, concluded that the alerting system matures early in development and the orienting and executive networks continue developing throughout childhood (*Pozuelos et al., 2014*). Later in life, alerting decreased with age, whereas the other two systems remained stable in adulthood (*Jennings et al., 2007*). Recent longitudinal studies revealed that alerting kept developing between the ages of 8 and 10, while orienting stabilized at 6 and executive stabilized at 7 (*Lewis et al., 2018*). Another longitudinal study in children with and without ADHD, who were assessed four times between the ages of 7 and 11, revealed improvement in all attention systems apart from orienting. Moreover, typically developing children showed superior executive functioning compared to children with ADHD (*Suades-González et al., 2017*).

Recent evidence supports the existence of five brain networks underpinning the three attention systems, one brain network for the alerting system, and two brain networks each for the orienting and the executive system (*Petersen and Posner, 2012*). The locus coeruleus and its projections to the frontal and parietal regions underpin alerting (*Aston-Jones and Cohen, 2005*; *Morrison and Foote, 1986*). The ventral attention network and the dorsal network, including the frontal eye fields and the IPS, underpin bottom-up orienting and top-down visuospatial functions, respectively (*Petersen and Posner, 2012*; *Corbetta et al., 2008*; *Hubbard et al., 2005*; *Lindner et al., 2010*; *Thompson et al., 2005*). The cingulo-opercular network and the frontoparietal network, mainly featuring the dorsolateral prefrontal cortex and the IPS, underpin the executive system (*Petersen and Posner, 2012*).

Several pharmacological studies have been conducted to date to examine the three attention systems. Pharmacological manipulation has shown that blocking norepinephrine decreases alerting (*Marrocco and Davidson, 1998*), whereas acetylcholine, via nicotine administration, reduces orienting performance (*Wignall and de Wit, 2011*). The administration of Adderall and Ritalin, which increase the concentration of dopamine and norepinephrine, improves executive control in individuals with ADHD. Animal work revealed the important role of glutamate and GABA, the brain's major excitatory and inhibitory neurotransmitters, in attention (*Pehrson et al., 2013*).

As discussed in the 'Introduction,' MRS has emerged as a valuable tool to investigate the role of neurochemicals across development. MRS studies on attention mainly compared individuals with and without ADHD symptoms. Children (8–12 y) with ADHD exhibit reduced concentration of GABA in the somatosensory/motor cortex compared to typically developing children (*Edden et al., 2012*), possibly

suggesting a deficit in short intracortical inhibition in ADHD. Glutamate was also investigated in the context of ADHD, where children with ADHD exhibited higher glutamate (*Moore et al., 2006*). Other MRS studies focused on N-acetylaspartate (NAA), a marker of neuronal integrity and health, yielding conflicting findings. For example, individuals with ADHD compared to controls exhibit below-normal NAA levels in some studies and above-normal levels in other studies (*Rüsch et al., 2010*; *Yang et al., 2010*). A meta-analysis identified a higher concentration of NAA in the prefrontal cortex in children with ADHD, but not in adults with ADHD, showing that the abnormal excess in NAA dropped linearly with increasing age, potentially resolving previous discrepancies between studies (*Aoki et al., 2013*).

The assessment of the three attention systems is typically done using the Attention Network Task (*Fan et al., 2002*; *Fan et al., 2003*; *Figure 1*), which allows comparing reaction times in different conditions (see below) to obtain for each participant the altering, orienting, and executive network indexes. The experimental difference scores in these cognitive tasks, which are based on the cognitive subtraction logic, are in alignment with the guidelines on how to compute these scores and are considered a standard and traditional way of quantifying these metrics. However, we acknowledge that recent psychometric work suggests that experimental difference scores from cognitive tasks can suffer from poor measurement properties, such as low reliability and unaccounted speed–accuracy interactions (*Draheim et al., 2019*).

However, the use of these metrics prevents the researcher from discerning the underlying processes of attention. To put it simply, if 14-year-old children exhibit superior executive systems scores compared to 6-year-old children, the researcher cannot determine whether this difference depends on the cognitive, perceptual, motor, decision processing, or a combination of these from the single reaction time measure alone.

Apart from regulating the processing of the attention system, the frontoparietal network is also engaged when we compare quantities in the context of nonsocial comparisons (e.g., which of two numbers is larger in magnitude, which of two dogs is taller) as well as in the context of social comparison (e.g., which of two women is more attractive). A classic effect in the literature that applies in the context of the frontoparietal network is the distance effect, which states that the closer two compared magnitudes (e.g. two numbers, physical attractiveness in two faces), the more difficult the comparison, and the greater the activity of this frontoparietal network (*Cohen Kadosh et al., 2005*; *Nieder and Dehaene, 2009*; *Kedia et al., 2014*; *Zacharopoulos et al., 2023*).

## Aims of the supplementary experiment

1. Given the crucial role of the frontoparietal network in regulating attention processing (see above) and for complementing the analyses of Task 1, we conducted a supplementary experiment that aimed to assess whether frontoparietal neurochemical concentration tracks and predicts the three diffusion parameters (i.e., mean drift rate, boundary separation, and visuomotor processing; for details, including multiple comparison corrections, see the 'Materials and methods' section) within each of the three attention networks separately. The alerting network was calculated by subtracting the diffusion parameter during the double-cue condition from the same diffusion parameter in the no-cue condition. The orienting network was calculated by subtracting the diffusion parameter during the spatial cue condition from the same diffusion parameter in the central cue condition. The executive network was calculated by subtracting the diffusion parameter during the congruent condition from the same diffusion parameter in the incongruent condition.

2. Given the crucial role of the frontoparietal network in regulating the distance effect (see above) and for complementing the analyses of Tasks 2–3, we examined whether frontoparietal neurochemical concentration tracks and predicts the three diffusion parameters (i.e., mean drift rate, boundary separation, and visuomotor processing; for details, see the 'Materials and methods' section) in response to the distance effect (in Tasks 2–3, and for completeness in response to the spatial-numerical association of response codes [SNARC] effect in Task 2). In Tasks 2–3, the distance effect was calculated by contrasting the high-distance vs. low-distance trials (in Task 2: far trials vs. near trials and in Task 3: 135-degree trials vs. 45-degree trials).

Regarding the first assessment analyses of the supplementary experiment, the only association that survived the FDR correction was the MFG NAA statistical model predicting boundary separation in response to the distance effect in Task 2, but this did not survive the neurochemical and neuroanatomical specificity. Regarding the longitudinal analyses of the supplementary

experiment, the only association that survived the FDR correction was the IPS glutamate statistical model predicting decision visuomotor processing within the alerting network in Task 1 ($\beta = -0.30$, t(143)=-2.55, $P_{BO} = 0.017$, CI = [-0.54–0.05]), and this result survived the neurochemical and neuro-anatomical specificity.

For completeness, however, for the attention network results as well as for the distance and SNARC effects, see *Supplementary file 4* (developmental effects), *Supplementary file 5* (neurochemical effects without controlling for behavioral scores during the first assessment when applicable), and *Supplementary file 6* (neurochemical effects with controlling for behavioral scores during the first assessment when applicable).

## Acknowledgements

The authors are grateful to all the participants and caretakers involved in this study, and to Malin I Karstens, Katarzyna Dabrowska, Laura Epton, Charlotte Hartwright, Ramona Kantschuster, Margherita Nulli, Claire Shuttleworth, and Anne Sokolich for their assistance in running this project. The authors are grateful to all Wellcome Centre for Integrative Neuroimaging (WIN) staff, in particular Nicola Filippini, Emily Hinson, Eniko Zsoldos, Caitlin O'Brien, Jon Campbell, Michael Sanders, Caroline Young, and David Parker. The Wellcome Centre for Integrative Neuroimaging (WIN) is supported by core funding from the Wellcome Trust (203139/Z/16/Z). This work was supported by the European Research Council (Learning&Achievement Grant 338065).

## Additional information

### Competing interests
Roi Cohen Kadosh: RCK serves on the scientific advisory boards of Neuroelectrics Inc, Innosphere Inc, and is the founder and shareholder of Cognite Neurotechnology Ltd. The other authors declare that no competing interests exist.

### Funding

| Funder | Grant reference number | Author |
| --- | --- | --- |
| Wellcome Trust | 203139/Z/16/Z | Roi Cohen Kadosh |
| European Research Council | 338065 | Roi Cohen Kadosh |

The funders had no role in study design, data collection and interpretation, or the decision to submit the work for publication. For the purpose of Open Access, the authors have applied a CC BY public copyright license to any Author Accepted Manuscript version arising from this submission.

### Author contributions
George Zacharopoulos, Conceptualization, Resources, Data curation, Software, Formal analysis, Validation, Investigation, Visualization, Methodology, Writing – original draft, Project administration, Writing – review and editing; Francesco Sella, Conceptualization, Data curation, Software, Supervision, Validation, Investigation, Visualization, Methodology, Project administration, Writing – review and editing; Uzay Emir, Resources, Data curation, Software, Supervision, Validation, Methodology, Writing – review and editing; Roi Cohen Kadosh, Conceptualization, Resources, Data curation, Supervision, Funding acquisition, Validation, Visualization, Methodology, Project administration, Writing – review and editing

### Author ORCIDs

George Zacharopoulos ⓘ https://orcid.org/0000-0003-0574-866X
Uzay Emir ⓘ https://orcid.org/0000-0001-5376-0431

### Ethics
Human subjects: Informed written consent was obtained from the primary caregiver and informed written assent was obtained from participants under the age of 16, according to approved institutional

guidelines. The study was approved by the University of Oxford's Medical Sciences Interdivisional Research Ethics Committee (MS-IDREC-C2_2015_016).

## Decision letter and Author response

Decision letter https://doi.org/10.7554/eLife.84086.sa1
Author response https://doi.org/10.7554/eLife.84086.sa2

## Additional files

### Supplementary files

• Supplementary file 1. Gender and mean age (standard deviation in parentheses) during the first (A1, top half) and the second (A2, bottom half) assessment.

• Supplementary file 2. Scatterplots depicting the association between chronological age (x-axis) and the diffusion parameters (y-axis) either during the first assessment (A1) or during the second assessment (A2) with a linear (yellow), quadratic (purple) and a cubic (green) fit, in the Attention Network Task (ANT), the Digit Comparison Task (DC), and the Mental Rotation Task (MRT). 1. Mean Drift Rate calculated across all the trials. 2. Boundary separation calculated across all the trials. 3. Non-decision time calculated across all the trials. 4. The three parameters of the alerting network (Task 1). The alerting network was calculated by subtracting the diffusion parameter during the double-cue condition from the same diffusion parameter in the no-cue condition. 5. The three parameters of the orienting network (Task 1). The orienting network was calculated by subtracting the diffusion parameter during the spatial cue condition from the same diffusion parameter in the central cue condition. 6. The three parameters of the executive network (Task 1). The executive network was calculated by subtracting the diffusion parameter during the congruent condition from the same diffusion parameter in the incongruent condition. 7. The three diffusion parameters of the distance effect obtained from contrasting "far" trials vs. "near" trials (Task 2). 8. The three diffusion parameters of the distance effect obtained from contrasting 135 degrees trials vs. 45 degrees trials (Task 3). 9. The three diffusion parameters of the SNARC effect obtained from contrasting low SNARC trials vs high SNARC trials (Task 2).

• Supplementary file 3. Multiple linear regressions with bootstrapping predicting overall visuomotor processing (A1: first assessment, A2: second assessment, β=the regression coefficient of the variable listed in the "Effect" column, df=degrees of freedom, T=t-statistic, $P_B$=Bootstrapped P-value, CI_L=lower bound of the confidence intervals obtained from bootstrapping, CI_U=upper bound of the confidence intervals obtained from bootstrapping) for Task 1 (Attention network task, top third), Task 2 (Digit comparison task, middle third), and Task 3 (Mental rotation task, bottom third).

• Supplementary file 4. Additional data regarding linear, quadratic, and cubic fits between diffusion parameters and chronological age (Task 1=Attention Network Task, Task 2=Digit Comparison Task, Task 3=Mental Rotation Task, v= mean drift rate, a=boundary separation, Ter=non-decision time, DF= degrees of freedom, rP=Pearson r, pP=p-value of rP, Spearman's rho=rS, pS=p-value of rS, l.a$R^2$=adjusted $R^2$ of the model featuring the intercept and the linear fit, q.a$R^2$=adjusted $R^2$ of the model featuring the intercept, the linear fit, and the quadratic fit, c.a$R^2$=adjusted $R^2$ of the model featuring the intercept, the linear fit the quadratic and the cubit fit). Regarding the column "Ord", here we run three models (i) M1 which featured the intercept and the linear fit, (ii) M2 which featured the intercept, the linear fit and the quadratic fit and (iii) M3 which featured the intercept, the linear fit, the quadratic fit and the cubic fit. Following that, we assigned three p-values, (i) the p-value of the linear fit from M1, (ii) the p-value of the quadratic fit from M2, and (iii) the p-value of the cubic fit from M3. If none of these p-values was less than .05, the "Ord" value is N/A. If only the p-value of the linear fit from M1 is significant then "Ord" is 1, if the p-value of the quadratic fit from M2 is significant but the p-value of the cubic fit from M3 is not significant then "Ord" is 2, and if the p-value of the cubic fit from M3 is significant then "Ord" is 3. Essentially, the "Ord" column indicates the highest order fit that significantly contributes to the data above and beyond the less higher order fit/s. The variables (both for the first and the second assessment) are sorted based on the Pearson's p-value of the first assessment.

• Supplementary file 5. Statistical results of neurochemicals in tracking behavioural performance. The first column denotes the task (i.e., Task 1, Task 2, or Task 3), the second column denotes the assessment where "A1" concerns the first assessment analyses corresponding to *Equation 3*, "A2" concerns the second assessment analyses corresponding to *Equation 4*, and "Prediction" concerns predicting behaviour during the second assessment based on neurochemicals during

the first assessment corresponding to *Equation 5*. The third column has three names separated by underscores, the first name corresponds to the region and the neurochemical used where IPS=intraparietal sulcus and MFG=middle frontal gyrus, and GLU=glutamate, GABA=gamma-Aminobutyric acid and NAA=N-acetylaspartate, the third name corresponds to the diffusion parameter that was used as the dependent variable, and the second name corresponds to the way each diffusion parameter was calculated where overall=the diffusion parameter was calculated across all the trials, AL=alerting network, OR=orienting network, EX=executive network, DISTANCE=the effect of distance, SNARC=the effect of SNARC. For the rest of the columns (df=degrees of freedom, β=standardized coefficient, $P_{BO}$=bootstrapped P-value), where β (column 5) and $P_{BO}$ (column 6) correspond to the neurochemical*age interaction and β (column 7) and $P_{BO}$ (column 8) correspond to the main effect of the neurochemical. The column "max_VIF" shows the maximum variance inflation factor assessing multicollinearity, and the column "SW" and "SW_P" shows the Shapiro-Wilk statistic and Shapiro-Wilk P-value, respectively, assessing the normality of the residuals in each model. The column "int_$R^2$" is the adjusted R-squared of the model, and the column "non_int_$R^2$" is the adjusted R-squared of the same model, but when omitting the interaction predictor and the column "delta_$R^2$" is the difference between "int_$R^2$" and "non_int_$R^2$". Of note, the variance inflation factor in the "Prediction" analyses was calculated, including all predictors apart from the predictor "age during the second assessment", as this predictor is bound to be positively correlated to the predictor "age during the first assessment".

• Supplementary file 6. Statistical results of SF6-eq1 (see below). To assess whether behaviour during the second assessment (i.e., A2) was predicted by neuroimaging measures during the first assessment (i.e., A1) while controlling for behaviour during the first assessment (i.e., A1), we employed SF6-eq1 as can be seen below which is a variant of (*Equation 5*) that additionally includes behaviour during the first assessment as can be seen below highlighted in bold. The first column denotes the task (i.e., Task 1, Task 2 or Task 3), the second column (i.e., "Prediction") concerns predicting behaviour during the second assessment based on neurochemicals during the first assessment. The third column has three names separated by underscores, the first name corresponds to the region and the neurochemical used where IPS=intraparietal sulcus and MFG=middle frontal gyrus, and GLU=glutamate, GABA=gamma-Aminobutyric acid and NAA=N-acetylaspartate, the third name corresponds to the diffusion parameter that was used as the dependent variable, and the second name corresponds to the way each diffusion parameter was calculated where overall=the diffusion parameter was calculated across all trials, AL=alerting network, OR=orienting network, EX=executive network, DISTANCE=the effect of distance, SNARC=the effect of SNARC. For the rest of the columns (df=degrees of freedom, β=standardized coefficient, $P_{BO}$=bootstrapped P-value), where β (column 5) and $P_{BO}$ (column 6) correspond to the neurochemical*age interaction and β (column 7) and $P_{BO}$ (column 8) correspond to the main effect of the neurochemical. The column "int_$R^2$" is the adjusted R-squared of the model, and the column "non_int_$R^2$" is the adjusted R-squared of the same model, but when omitting the interaction predictor and the column "delta_$R^2$" is the difference between "int_$R^2$" and "non_int_$R^2$".

$$behavioural\ score\ (A2)\ \sim\ neurotransmitter\ (A1)\ +\ age\ (A1)\ +\ neurotransmitter\ (A1)\ *\ age\ (A1)\ +$$

$$control\ behavioural\ score\ 1\ (A1)\ +\ control\ behavioural\ score\ 2\ (A1)\ +\ control\ behavioural\ score\ 1\ (A2)\ +$$

$$control\ behavioural\ score\ 2\ (A2)\ +\ \mathbf{behavioural\ score\ (A1)}\ +\ age\ (A2)$$

• Supplementary file 7. Moderated mediation results. At the start of each entry, the dependent variable (Y), the independent variable (X), the mediator (M), and the moderator (W) are defined and highlighted in bold. 1. IPS GABA moderated mediation model with visuomotor connectivity as the mediator. 2. IPS glutamate moderated mediation model with visuomotor connectivity as the mediator. 3. IPS GABA moderated mediation model with visuomotor processing as the mediator. 4. IPS glutamate moderated mediation model with visuomotor processing as the mediator.

• Supplementary file 8. Age moderated the relationship between diffusion parameters and fluid intelligence (CI_L= 95% lower bound confidence interval, CI_U= 95% upper bound confidence interval). We did not control for the other two diffusion parameters in these analyses.

• Supplementary file 9. Mean accuracy, mean reaction time (RT) and associated standard deviations (SD) in each of the three tasks (indicated in the first column).

• Supplementary file 10. Correlations between the diffusion parameters (v=mean drift rate, a=boundary separation, Ter=non-decision time) within and between tasks. As expected, positive correlations of the three diffusion parameters were obtained across the three different tasks

assessing attention (Task 1, ANT), digit comparison (Task 2, DC) and mental rotation (Task 3, MRT).

• Supplementary file 11. Statistical results using the same statistical model as in *Supplementary file 3* (multiple linear regressions with bootstrapping predicting overall visuomotor processing during the first and the second assessment) but using the raw neurochemical concentration values (A1: first assessment, A2: second assessment, β=the regression coefficient of the variable listed in the "Effect" column, df=degrees of freedom, T=t-statistic, $P_B$=Bootstrapped P-value, CI_L=lower bound of the confidence intervals obtained from bootstrapping, CI_U=upper bound of the confidence intervals obtained from bootstrapping) for Task 1 (Attention network task, top third), Task 2 (Digit comparison task, middle third), and Task 3 (Mental rotation task, bottom third).

• Supplementary file 12. Statistical results using the same statistical model as in *Supplementary file 3* (multiple linear regressions with bootstrapping predicting overall visuomotor processing during the first and the second assessment) but using the relative to total creatine neurochemical concentration values (A1: first assessment, A2: second assessment, β=the regression coefficient of the variable listed in the "Effect" column, df=degrees of freedom, T=t-statistic, $P_B$=Bootstrapped P-value, CI_L=lower bound of the confidence intervals obtained from bootstrapping, CI_U=upper bound of the confidence intervals obtained from bootstrapping) for Task 1 (Attention network task, top third), Task 2 (Digit comparison task, middle third), and Task 3 (Mental rotation task, bottom third).

• Supplementary file 13. Statistical results using the same statistical model as in *Supplementary file 3* (multiple linear regressions with bootstrapping predicting overall visuomotor processing during the first and the second assessment) but using the T2 corrected (*Equation 2*) neurochemical concentration values (A1: first assessment, A2: second assessment, β=the regression coefficient of the variable listed in the "Effect" column, df=degrees of freedom, T=t-statistic, $P_B$=Bootstrapped P-value, CI_L=lower bound of the confidence intervals obtained from bootstrapping, CI_U=upper bound of the confidence intervals obtained from bootstrapping) for Task 1 (Attention network task, top third), Task 2 (Digit comparison task, middle third), and Task 3 (Mental rotation task, bottom third).

• Supplementary file 14. Complementary factor analysis results. We conducted exploratory factor analyses by adding as input the non-decision time parameters of the three tasks (i.e., three variables as input) after controlling for age, and only one factor was extracted (i.e., eigenvalue>1). As can be seen in the table below, this factor (extraction method: Principal Component Analysis, rotation method: none) was consistently positively related to the non-decision time (Ter) of each of the three tasks.

• Supplementary file 15. Statistical results of multiple linear regressions with bootstrapping during the first assessment (β=the regression coefficient of the variable listed in the "Effect" column, $P_B$=Bootstrapped P-value, CI_L=lower bound of the confidence intervals obtained from bootstrapping, CI_U=upper bound of the confidence intervals obtained from bootstrapping). These additional analyses support that both predictors (i.e., IPS glutamate*age and IPS GABA*age) were independently significant in tracking task performance (non-decision time in Task 1, Task 2 and when Task 1-3 were combined) even when both were added in the same multiple regression model.

• Supplementary file 16. To address the possibility that non-linear effects of age could impact the main analyses reported in the main text, we conducted additional analyses where we compared the model in SF16-eq1 to the model in SF16-eq2 using an R-change ANOVA test, which was significant.

• MDAR checklist

• Source data 1. MRS, resting fMRI and behavioural data.

### Data availability

All quantitative data (i.e., behavioural, neuroimaging) underlying the results can be found in *Source data 1*.

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
