## [Editor Report]

This important study combines behavioral and imaging experiments to understand how levels of important brain chemicals shape the processing of information in the brain in children and young adults. The sample size and data quality are outstanding and some of the data are convincing. However, there are important caveats for some of the results, as discussed by the authors, and future replication will be important to fully substantiate the findings. This work will be of interest to neuroscientists, psychologists, and neuroimaging researchers investigating the developing brain in health and disease.

---

## [Decision Letter]

**Decision letter after peer review:**

Thank you for submitting your article "Dissecting the chain of information processing and its interplay with neurochemicals and fluid intelligence across development" for consideration by *eLife*. Your article has been reviewed by 3 peer reviewers, and the evaluation has been overseen by a Reviewing Editor and Chris Baker as the Senior Editor. The following individuals involved in the review of your submission have agreed to reveal their identity: Georg Oeltzschner (Reviewer #1); Reuben Rideaux (Reviewer #2); Alexander Weigard (Reviewer #3).

Essential revisions:

While the reviewers highlight many strengths of the work including the extensive nature of the dataset, they raise some substantial concerns that significantly weaken the interpretability of the results. There are two major areas of concern.

1) There is a need for more transparency regarding the analytic decisions made in the manuscript and greater justification for those decisions. While there are aspects of compelling robustness (e.g., large sample size, test re-test replication), these are weakened by factors suggesting unreliability (e.g., multiple comparisons, liberal α values, selective reporting). All three reviewers have provided clear and concrete suggestions for improving the reporting of the methods and analyses and the implicit assumptions made in those decisions.

R1's comment regarding the choice of metabolite quantification methods provides a good example. There are many different methods for MRS processing and quantification, and each decision in the pipeline produces a different set of estimates at the end. While there may be justifiable reasons for using the analysis presented in the manuscript, these decisions are currently opaque, which prevents an assessment of their legitimacy. Reporting the results of using alternative processing/quantification methods (e.g., those previously used by the group on this dataset) would give an indication of the robustness of the findings.

2) Reviewer 3 highlights a number of concerns about the interpretation and use of the DDM parameters and the associated mediation analyses. In particular, they highlight the weak conceptual link between the t0 parameter and intelligence in prior work and are concerned about the α value of 0.10 that seems to have been chosen post hoc in the mediation analyses. The claims with regard to both of these aspects of the manuscript may need to be weakened.

In addition, you should include greater consideration of age trajectories of MRS-derived metabolite estimates during childhood and early adulthood.

*Reviewer #1 (Recommendations for the authors):*

1. I have found the MRS methodological section to be largely inadequate at this stage – there is a considerable lack of detail regarding acquisition and data analysis, as well as some unresolved confusion, particularly with respect to the exact corrections being made. In addition, the terminology is a little ambiguous and potentially misleading.

a. "Absolute neurochemical concentrations were then scaled based on the structural properties of the selected regions and based on the predefined values shown in MRS-eq1 and MRS-eq2 (see below) (84); these predefined scaling values were therefore determined before the data collection." This sentence is very troubling and unclear, particularly the part "these predefined scaling values were therefore determined before the data collection". Tissue correction of water-scaled MRS levels is done on a subject-specific basis, i.e. by co-registering the MRS voxel to the structural image, and determining the (subject-specific) fractional tissue volumes for GM/WM/CSF. You describe how you have implemented precisely this routine – and it is, therefore, difficult for me to understand why you state "the scaling values were determined before the data collection". Does this mean that co-registration has only been done once, and the segmentation is then assumed to be valid (and identical) for all subjects? This would counteract the benefits of individual tissue and relaxation correction. Please clarify, and if necessary, please repeat the analysis with subject-specific tissue and relaxation correction.

b. Generally, the term "absolute concentrations" is advised against, because absolute quantification relies on a lot of assumptions. As stated in a recent consensus paper on MRS terminology: "The term absolute quantification was originally coined for the conversion to standard concentration units, where the conversion includes all corrections needed based on measurements and calibrations for the currently reported case to contrast with relative quantification. However, this narrow meaning is an idealization that normally cannot be realized in practice, and "absolute quantitation" today is also used for plain conversion to standard units with calibrations from earlier studies or literature values, but this usage is not encouraged" (https://onlinelibrary.wiley.com/doi/10.1002/nbm.4347).

c. In this paper, you are using the term "absolute neurochemical concentrations" equivalent to un-corrected metabolite estimates with respect to the internal tissue water, as output by LCModel, starting with the sentence "Absolute neurochemical concentrations were extracted from the spectra using a water signal as an internal concentration reference". This is unclear at best, and misleading at worst, and should be changed to avoid confusion. You need to explicitly state that you start with the "raw" metabolite-to-water amplitude ratios that LCModel returns, and describe in greater detail how they are produced. By default, LCModel assumes pure white matter as water concentration reference, and already performs an extremely basic relaxation correction using the parameters WCONC, ATTH2O, and ATTMET. Please describe how all of these parameters were set in the control file.

d. Likewise, the term "scaling" is not often used to describe the procedure of tissue and relaxation corrections. Please use the appropriate term "correction".

e. Quality control. You mention that cases with CRLB>50% or SNR or concentration value beyond 3 SD were excluded from further analyses. How many cases were excluded according to each criterion? Is Figure 1 showing the spectra before or after exclusion? If a GABA CRLB was >50%, was the entire spectrum discarded, or did you keep the Glu value (which was likely never >50%)?

f. There is no mention of the MRI field strength, vendor, or scanner type. The MRS paragraph gives it away with the sentence "Simulations were performed using the same RF pulses and sequence timings as in the 3T system described above", but the system is not described above (unlike in the other papers published with identical methods sections). Please also provide more information about the MRspa process, specifically about the processing steps that were being taken (coil combination, alignment of individual transients, etc.).

g. The experimental description is a little misleading. If my calculations are correct, the pure measurement time per voxel should have been roughly 2 minutes, but you state that sequence planning and shimming took 10-15 minutes. This is because you don't mention the acquisition of the water T2 series in every participant, except for the one paragraph during data processing.

h. Please show a complete LCModel fit, not just a single GABA and Glu model. This should include the spectrum itself (primarily to judge single-spectrum SNR), the fit, the residual, the baseline estimation, and the individual contributions from all basis functions (for example, as a stacked plot).

i. To illustrate the variability of the GABA and Glu fits, you could consider showing their mean +/- SD as well, not just a single example (or even all of them on top of each other).

2. Introduction

a. "In other words, do the associations between frontoparietal neurochemicals and visuomotor processing is replicated across different tasks such as numerical processing (Figure 1, Task 2) and mental rotation (Figure 1, Task 3)?"- This should probably read "replicate across different tasks".

b. Reference (34) in "For example, it is known that neurochemical changes may precede anatomic changes (34)" is a study on brain tumors. It is a bit of a stretch to derive from this a "unique predictive ability of MRS" to discern how neurochemicals shape the developmental trajectory of neurobiological processes. I would suggest removing this particular reference – or adding ones that are more relevant to development and metabolite age trajectories (e.g., https://pubmed.ncbi.nlm.nih.gov/34061022/).

3. Figures

a. The caption of Figure 5 includes descriptions of panels A-D, but only shows three panels A-C.

*Reviewer #2 (Recommendations for the authors):*

Line 105: "…explained current and predicted future developmental changes in numerical cognition…" – I think it would be more accurate to state that ref 26 found a relationship between GABA/Glu concentration and mathematical achievement.

Figure 1a: Are these examples of anatomical T1-weighted images? I think more information on what is being shown in this panel would be useful in the figure caption. Also, the contrast seems quite low, these images may be easier to interpret if the luminance is increased.

Figure 1: The resolution of this figure is very low, which makes it difficult to assess it. For example, the text below the attention network task is barely legible.

Line 129: "In other words, do the associations between frontoparietal neurochemicals and visuomotor processing is replicated…" – a typo here.

Line 134: "A more optimal and informative way to address these questions is by using…" This is a strange way to begin a paragraph given that you have not discussed an alternative – less optimal – way of addressing those questions in the previous paragraph. Consider rewriting for clarity.

Line 139: "Therefore, it is possible to utilize and extend this unique predictive ability of MRS to discern which of these key neurobiological processes (cognitive, decision, visuomotor) are shaped by highly specialized neurochemical concentration across development." – This language seems to imply that finding a correlation between MRS detected metabolite concentrations and task performance is indicative of a causal relationship. I think it's important to be clear that correlations, even those that are predictive of future states, are not necessarily indicative of causal relationships.

Supplementary File 2: This file could be made clearer. Including more than two plots on each page, and grouping them according to either task or model parameters would allow for easier comparison. Some of the y-axis labels have typos, e.g., DISTANCEFARMINUSCLOSE, which I think are probably due to the way MATLAB treats underscores in text. Perhaps try using different clearer or correct these issues in a separate figure editing package. I think it would be useful to include an explanation of all the terms (e.g., alerting, orienting, executive, etc.) at the top of the file.

A large number of correlations were tested in this study. I appreciate that the authors performed an FDR correction and tested the test-retest reliability with a 1.5-year gap. I think that it would also be useful to see the distribution of all the correlation values, i.e., in a histogram, to get an idea of how distinguishable those that were categorized as meaningful are from those that were categorized as not meaningful.

I understand the choice of colour schemes for the scatterplots in the main text (blue and red for excitation and inhibition, respectively), however, the individual datapoints are difficult to parse between groups because they are all the same colour and semi-transparent. To improve the clarity of the plots, you could use different colours for each group, or something similar. Also, making the dots smaller will reduce the overlap and probably make them easier to distinguish from one another.

Figures 2-4: Please indicate the units of the x and y labels. Also, please indicate in the figure caption whether higher scores in visuomotor processing are associated with better or worse task performance.

Figures 2-4: "For visualization purposes, we did not control for boundary separation and mean drift rate when plotting these panels." – I don't understand how not controlling for the variability helps visualization, could the authors please clarify?

The authors find a correlation between IPS GABA and Glu and visuomotor processing in three different tasks. It would be interesting to test whether there is independent variability explained by the neurotransmitters in each task or not. For example, how correlated are the visuomotor processing scores between tasks, and if you control for the variability in one/two tasks, is there still a correlation between GABA/Glu and visuomotor processing in the third task?

Line 367: "These findings suggest that the relationship between IPS glutamate and GABA and visuomotor processing across development is task-independent." – This is related to the previous comment. This may have very little to do with GABA/Glu and more to do with the reliability of measuring the same latent construct with these three tasks. I think that the analysis suggested above could help to test this and to put this assertion into a better context.

A recent study found evidence for a brain-wide positive relationship between GABA and Glu concentration in adults (Rideaux et al., 2022, NeuroImage). Thus, one might predict that if GABA were positively correlated with task performance, then Glu would also be positively correlated. By contrast, in the current study it appears that for the majority of correlations between GABA and task performance, there appears to be an inverse relationship with Glu. This seems quite interesting, could the authors speak to this? For example, does the relationship between GABA and Glu change with age?

The authors found relationships between IPS GABA/Glu and task performance, but not MFG GABA/Glu. This could be due to poorer signal-to-noise in the MFG measurements. Can the authors test this possibility by comparing measures of signal quality between the voxel locations, i.e., FWHM, CRLB, SNR, and fitting residuals?

Line 310: "Since we identified a strong and task-independent developmental effect between IPS neurochemicals and visuomotor processing…" – By what categorization method is this effect strong? Most of the β values seem relatively weak (e.g., between 0.15-0.20). Perhaps the authors mean reliable?

Line 447: "…through the visuomotor resting-state connectivity depending on the individual's developmental stage." – This claim relies on both positive results, e.g., evidence of a relationship at some ages, and negative results, e.g., evidence of no relationship at other ages. However, the latter can only be established by directly testing the likelihood of the null hypothesis, e.g., with Bayesian statistics.

The paragraph starting at line 461: Related to the above point, several explanations are provided for why the relationship was only found for older participants. Another explanation is that the data may have been noisier for younger participants which led to a failure to detect the relationship.

Please provide a plot showing the positioning of the voxels, ideally with the degree of overlap between participants.

Please report the following MRS information: spectral width, number of data points acquired, and number of water-unsuppressed averages.

Please report the phase and frequency alignment method used on subspectra prior to averaging.

Please report the test-retest reliability of the GABA and Glu concentration estimates and compare them with previously reported test-retest reliability studies, e.g., Saleh et al., 2016, Magn. Reson. Mater. Phys. Biol. Med.

It is reported in the Methods that metabolite estimates were rejected as outliers prior to performing tissue correction. I think that tissue correction should be performed prior to outlier rejection.

Line 682: "…as this helped to assess the whether the results.." – typo.

Line 713: "…but this did not survive the neurochemical and neurotransmitter specificity…" – What is the difference between neurochemical and neurotransmitter specificity?

Please report the order in which participants performed the 3 tasks (was it always the same, randomized, or counterbalanced?). Related to this, FDR correct was performed on the attention network task correlations, but not on the subsequent "replication" correlations with the other two tasks (digit comparison and mental rotation). Could the authors please specify whether analyses were performed on the attention network task, prior to analysing the correlations between GABA/Glu and the other two tasks?

Line 760: "Since we obtained the same pattern of findings regarding visuomotor processing across Task 1-3 which was one of our earlier aims was addressed…" – typo.

Line 764: "zscored" – hyphenate.

[Editors' note: further revisions were suggested prior to acceptance, as described below.]

Thank you for resubmitting your work entitled "Dissecting the chain of information processing and its interplay with neurochemicals and fluid intelligence across development" for further consideration by *eLife*. Your revised article has been evaluated by two of the original reviewers and by Chris Baker as Senior/Reviewing Editor.

The manuscript has been improved but there are some remaining issues that need to be addressed, as outlined below.

While the reviewers appreciate the detailed responses to the initial comments, there remain a number of specific issues that you need to address in a revision (see detailed comments from the two reviewers below). In particular, both reviewers think you need to be more open about the limitations of the data and analyses and temper some of the claims. This applies to issues such as correlations between GABA/Glu and task performance (R1), analyses of signal quality (R1), and mediation analyses (R2).

Additional analyses and careful revisions will be required for a positive decision on the manuscript.

*Reviewer #2 (Recommendations for the authors):*

The revised manuscript is an improvement on the original and the authors have addressed most of my previous reviews. However, there remain a number of outstanding issues, which will need to be addressed.

There are both red and pink data points in Figures 2-6, yet these don't appear to be labelled or described. Please fix this if it is a visualization error, otherwise please explain the distinction.

The authors misunderstood my previous request. I wasn't asking whether task performance is correlated. I had asked if there were independent relationships between GABA and task performance and Glu and task performance. For example, if you partial out the variability of GABA from Glu, does the relationship between Glu and task performance disappear? If so, this would make it difficult to infer specific associations between GABA/Glu and these tasks. In particular, Glu and GABA measurements tend to share variance, due to both metabolic (e.g., synthesis cycles) and non-metabolic factors (e.g., signal quality, frequency drift, subject age). If task performance is associated with this shared pool of GABA-Glu variance, then it is difficult to say which neurotransmitter is involved or (worse) if the task isn't simply correlated with the variance associated with non-metabolic factors. I think this is an important point and should be addressed.

The authors found that the signal quality in MFG was significantly poorer than that in IPS, as indicated by multiple indicators of signal quality. Please report this in the main text of the manuscript and acknowledge that this could explain why a relationship was not detected between MFG GABA/Glu and task performance. I appreciate that the authors feel confident that this is not the explanation; however, I think it's important that readers be made aware of the significantly lower signal quality of the MFG spectra.

The authors indicate that they are confident that the MRS measurements from all age groups have sufficient signal quality to detect meaningful relationships with their additional measures because they have excluded the datasets with poorest signal quality. This point is also offered in R2.R11. Matching the datasets in terms of their minimum signal quality is not sufficient to show that signal quality differences cannot explain the failure to detect relationships with other measurements. To make this claim, the authors need to demonstrate that the average signal quality is similar between groups. Please compare the average signal quality between groups, as was done between IPS and MFG in R2.R11. and report whether there are significant differences in the main text of the manuscript. If the average signal quality is significantly lower in groups where relationships were not detected, please acknowledge that lower signal quality may explain the failure to detect these relationships.

Please increase the luminance of the T1-weighted images to make the structure more visible and provide axial, sagittal, and coronal views for each voxel.

I think that the authors have misinterpreted my request – I agree that LCModel is an established method of metabolite quantification. I requested that the authors show the test-retest reliability of the GABA/Glu measurements for two reasons: (1) this is useful comparative information regarding MRS measurements and will be of interest to a broad range of readers and (2) because it provides the reader with an indication of the dynamic reliability/consistency of these measurements, which is influenced by many factors other than the quantification algorithm applied. Thus, could the authors please provide the test-retest reliability of the measurements (not simply the p-values for the correlation between them)?

*Reviewer #3 (Recommendations for the authors):*

In my view, the authors were moderately responsive to reviewer comments. I appreciate the authors' clarifications of prior work on DDM parameters and intelligence, the motivation for the supplemental experiment, and the resting state fMRI analysis procedures. I also think that the addition of the parameter correlation tables and the supplemental EFA increase confidence that the Ter parameter being measured is a task-general construct. I cannot speak as to whether the additional details about the MRS aspects of the study would impact the other reviewers' concerns, but it does seem like there is now greater transparency about the MRS procedures. I still disagree with the choice to investigate DDM parameters across the three tasks as three manifest variables rather than a single latent variable and the choice to visualize effects by binning ages rather than showing age as a continuous factor, but in both cases, I could see why there are reasonable arguments for the choices the authors made. However, I still have several concerns that I do not think were adequately addressed.

Simply showing the fits of nonlinear models to age differences in DDM parameter values in supplemental materials does not really address the possibility that non-linear effects of age could impact the main analyses reported in the paper, in which linear age appears to moderate the associations between DDM parameter values and MRS measures. There appears to be clear nonlinearity in relationships between age and Ter, the parameter that received the greatest focus. Unless there is a clear reason why sensitivity analyses that include nonlinear age terms would not be possible, such sensitivity analyses would be very valuable for determining whether the primary findings of the study are robust to these considerations.

The evidence from the mediation analysis is still concerningly weak. The authors now acknowledge the post hoc nature of the analyses and the liberal, uncorrected α level (although the other reason for concern, the inconsistency of effects across age groups, does not seem to be acknowledged as a limitation of the evidence for these effects). However, the mediation effects are still framed as a central claim of the paper. I think it makes sense to report the mediation results for completeness and to describe them as preliminary findings that could be replicated, but given the weakness of the evidence I don't think it makes sense to describe these results as a central finding. For example, the statement in the abstract that "We showed that fluid intelligence performance is explained by IPS GABA and glutamate and is mediated by visuomotor processing" seems like much too strong of a claim given the available evidence and the inconsistency in the details of the mediation effects across age groups. The age moderation effects of associations between DDM parameters and MRS measures that are reported seem to be robust and interesting in their own right, which makes me question the need to highlight a mediation analysis that provides evidence that, in my view, can be thought of as preliminary at best.

I appreciate the authors' addition of key details about how the DDM parameters were estimated. It seems like several of these details would be appropriate to mention as limitations in the discussion, including the relatively low number of trials and the high accuracy rate of the tasks, both of which can negatively impact the reliability of parameter estimates. However, a parameter recovery study would greatly increase confidence in the assumption that these features of the task data did not impede the accurate estimation of DDM parameters.

---

## [Author Response]

Essential revisions:Reviewer #1 (Recommendations for the authors):1. I have found the MRS methodological section to be largely inadequate at this stage – there is a considerable lack of detail regarding acquisition and data analysis, as well as some unresolved confusion, particularly with respect to the exact corrections being made. In addition, the terminology is a little ambiguous and potentially misleading.a. "Absolute neurochemical concentrations were then scaled based on the structural properties of the selected regions and based on the predefined values shown in MRS-eq1 and MRS-eq2 (see below) (84); these predefined scaling values were therefore determined before the data collection." This sentence is very troubling and unclear, particularly the part "these predefined scaling values were therefore determined before the data collection". Tissue correction of water-scaled MRS levels is done on a subject-specific basis, i.e. by co-registering the MRS voxel to the structural image, and determining the (subject-specific) fractional tissue volumes for GM/WM/CSF. You describe how you have implemented precisely this routine – and it is, therefore, difficult for me to understand why you state "the scaling values were determined before the data collection". Does this mean that co-registration has only been done once, and the segmentation is then assumed to be valid (and identical) for all subjects? This would counteract the benefits of individual tissue and relaxation correction. Please clarify, and if necessary, please repeat the analysis with subject-specific tissue and relaxation correction.

We have now added additional text to avoid any confusion. The text now reads, “By predefined correction values, we specifically refer to the values 43300, 35880, and 55556, which were the water concentrations in mmol/L for GM, WM, and CSF (in MRS-eq1). On the other hand, the GM, WM and CSF fractions (MRS-eq1) and T2 of tissue water value (MRS-eq2) were not determined prior to data collection. Instead, these values were calculated in a subject-specific manner as described below.” (page: 33, paragraph: 3, line: 20)

b. Generally, the term "absolute concentrations" is advised against, because absolute quantification relies on a lot of assumptions. As stated in a recent consensus paper on MRS terminology: "The term absolute quantification was originally coined for the conversion to standard concentration units, where the conversion includes all corrections needed based on measurements and calibrations for the currently reported case to contrast with relative quantification. However, this narrow meaning is an idealization that normally cannot be realized in practice, and "absolute quantitation" today is also used for plain conversion to standard units with calibrations from earlier studies or literature values, but this usage is not encouraged" (https://onlinelibrary.wiley.com/doi/10.1002/nbm.4347).c. In this paper, you are using the term "absolute neurochemical concentrations" equivalent to un-corrected metabolite estimates with respect to the internal tissue water, as output by LCModel, starting with the sentence "Absolute neurochemical concentrations were extracted from the spectra using a water signal as an internal concentration reference". This is unclear at best, and misleading at worst, and should be changed to avoid confusion. You need to explicitly state that you start with the "raw" metabolite-to-water amplitude ratios that LCModel returns, and describe in greater detail how they are produced. By default, LCModel assumes pure white matter as water concentration reference, and already performs an extremely basic relaxation correction using the parameters WCONC, ATTH2O, and ATTMET. Please describe how all of these parameters were set in the control file.d. Likewise, the term "scaling" is not often used to describe the procedure of tissue and relaxation corrections. Please use the appropriate term "correction".

Following the reviewer’s advice, we have now replaced the term “scaling” with the term “correction” and also replaced the term “absolute” with the term “raw”, and we clearly define the term raw. The text reads, “Raw neurochemical concentrations (i.e., metabolite-to-water amplitude ratios) were extracted from the spectra using a water signal as an internal concentration reference. We used the lcmodel default output with the following parameters: ATTMET=32, ATTH2O=1, WCONC=55555, PPMSHF=0.0, PPMEND=0.5, PPMST=4.2, DKNTMN=0.25, RFWHM=2.5, SDDEGP=5.00, SDDEGZ=5.00, DOWS=TRUE, DOECC=FALSE, DELTAT=0.0001666, NUNFIL=2048, HZPPPM=123.2571, DEGPPM=0, DEGZER=0.” (page: 32, paragraph: 3, line: 22)

e. Quality control. You mention that cases with CRLB>50% or SNR or concentration value beyond 3 SD were excluded from further analyses. How many cases were excluded according to each criterion? Is Figure 1 showing the spectra before or after exclusion? If a GABA CRLB was >50%, was the entire spectrum discarded, or did you keep the Glu value (which was likely never >50%)?

The number of cases excluded according to each criterion was as follows (T1=first assessment, T2=second assessment, tc=tissue corrected, t2c=T2 tissue corrected). If a GABA CRLB was >50%, we did not exclude the entire spectrum (i.e., we kept the glutamate value if the glutamate value survived the exclusion criteria and vice versa). Figure 1 shows the spectra after the exclusion.

**Author response table 1. sa2table1:** 

	Concentration	SNR	CRLB
T1 MFG GABA tc	0	8	10
T1 MFG glutamate tc	1	8	2
T1 IPS GABA tc	1	3	2
T1 IPS glutamate tc	1	3	0
T1 MFG GABA t2c	2	8	10
T1 MFG glutamate t2c	4	8	2
T1 IPS GABA t2c	1	3	2
T1 IPS glutamate t2c	1	3	0
T2 MFG GABA tc	0	5	7
T2 MFG glutamate tc	2	5	2
T2 IPS GABA tc	0	2	2
T2 IPS glutamate tc	0	2	2
T2 MFG GABA t2c	0	5	7
T2 MFG glutamate t2c	2	5	2
T2 IPS GABA t2c	0	2	2
T2 IPS glutamate t2c	0	2	2

f. There is no mention of the MRI field strength, vendor, or scanner type. The MRS paragraph gives it away with the sentence "Simulations were performed using the same RF pulses and sequence timings as in the 3T system described above", but the system is not described above (unlike in the other papers published with identical methods sections). Please also provide more information about the MRspa process, specifically about the processing steps that were being taken (coil combination, alignment of individual transients, etc.).

We now provide all of this information, and the text now reads, “All MRI data were acquired using a 3T Siemens MAGNETOM Prisma MRI System equipped with a 32 channel receive-only head coil. Anatomical high-resolution T1-weighted scans were acquired consisting of 192 slices, repetition time (TR)=1,900ms; echo time (TE)=3.97ms; voxel size = 1 × 1 × 1 mm). Simulations were performed using the same RF pulses and sequence timings as in the 3T system described above (Terpstra et al., 2016). The MRspa pre-processing steps included the following: 1) cross-correlation method to minimise the frequency difference between single shot MRS data, 2) least square method to minimise the phase difference between single shot MRS data, 3) Eddy Current Correction (ECC) was used to correct for both ECC based on the phase of water reference scan and 4) save the averaged spectrum.” (page: 31, paragraph: 2, line: 17)

g. The experimental description is a little misleading. If my calculations are correct, the pure measurement time per voxel should have been roughly 2 minutes, but you state that sequence planning and shimming took 10-15 minutes. This is because you don't mention the acquisition of the water T2 series in every participant, except for the one paragraph during data processing.

We have now added additional information to avoid confusion, and the text now reads “Acquisition time per voxel was 10–15 minutes, including manual voxel placement, sequence planning, shimming, potential readjustments if necessary and the acquisition of the water T2 series (see below).” (page: 32, paragraph: 2, line: 12)

h. Please show a complete LCModel fit, not just a single GABA and Glu model. This should include the spectrum itself (primarily to judge single-spectrum SNR), the fit, the residual, the baseline estimation, and the individual contributions from all basis functions (for example, as a stacked plot).

Please find the stacked plot below, which we have now included in the Supplement, and the text now states, “For a stacked plot depicting the spectrum, the fit, the residual, the baseline estimation, and the individual contributions from all basis functions, see Figure 1—figure supplement 1.” (page: 33, paragraph: 1, line: 2)

i. To illustrate the variability of the GABA and Glu fits, you could consider showing their mean +/- SD as well, not just a single example (or even all of them on top of each other).

We have now generated such plots and added them in Figure 1.

2. Introductiona. "In other words, do the associations between frontoparietal neurochemicals and visuomotor processing is replicated across different tasks such as numerical processing (Figure 1, Task 2) and mental rotation (Figure 1, Task 3)?"- This should probably read "replicate across different tasks".

We have now amended that sentence and the text now reads “In other words, do the associations between frontoparietal neurochemicals and visuomotor processing replicate across different tasks such as numerical processing (Figure 1, Task 2) and mental rotation (Figure 1, Task 3)?” (page: 6, paragraph: 1, line: 5)

b. Reference (34) in "For example, it is known that neurochemical changes may precede anatomic changes (34)" is a study on brain tumors. It is a bit of a stretch to derive from this a "unique predictive ability of MRS" to discern how neurochemicals shape the developmental trajectory of neurobiological processes. I would suggest removing this particular reference – or adding ones that are more relevant to development and metabolite age trajectories (e.g., https://pubmed.ncbi.nlm.nih.gov/34061022/).

We thank the reviewer for this suggestion. We have now deleted the following sentence and corresponding reference (i.e., “For example, it is known that neurochemical changes may precede anatomic changes (Chuang et al., 2016).”), and we added the suggested reference in this sentence “Therefore, it is possible to utilise and extend this unique predictive ability of MRS to discern which of these key neurobiological processes (cognitive, decision, visuomotor) are associated with highly specialised neurochemical concentrations across development (Porges et al., 2021).” (page: 6, paragraph: 2, line: 13)

3. Figuresa. The caption of Figure 5 includes descriptions of panels A-D, but only shows three panels A-C.

We have now deleted the following redundant text from Figure 5 caption.

Reviewer #2 (Recommendations for the authors):1. Line 105: "…explained current and predicted future developmental changes in numerical cognition…" – I think it would be more accurate to state that ref 26 found a relationship between GABA/Glu concentration and mathematical achievement.

We thank the reviewer for noticing this error, as we predicted future mathematical achievement [as can be seen in Figure 3. Predicting MA at Time 2 from the interaction between age neurotransmitter concentration at Time 1. (Zacharopoulos, Sella, Cohen Kadosh, et al., 2021)] but not a change. We have now amended the text, and the Introduction now reads, “In a similar vein, a recent magnetic resonance imaging (MRS) study revealed that glutamate and GABA profiles within the classic frontoparietal network regions, namely, the intraparietal sulcus (IPS, see Figure 1—figure supplement 2B) and the middle frontal gyrus (MFG, see Figure 1—figure supplement 2A), explained current and predicted future mathematical achievement levels (Zacharopoulos, Sella, Cohen Kadosh, et al., 2021).” (page: 5, paragraph: 1, line: 1)

2. Figure 1a: Are these examples of anatomical T1-weighted images? I think more information on what is being shown in this panel would be useful in the figure caption. Also, the contrast seems quite low, these images may be easier to interpret if the luminance is increased.

We have now increased the luminance as suggested and have provided additional information. The figure caption now states, **“**(A) Examples of T1-weighted images collected during the first assessment (A1) and the second assessment (A2, ~1.5 years later) in each of the five age groups (at A1: 6-year-olds, 10-year-olds, 14-year-olds, 16-year-olds, and 18+-year-olds).”.

3. Figure 1: The resolution of this figure is very low, which makes it difficult to assess it. For example, the text below the attention network task is barely legible.

We have now substantially increased the resolution of the figure so that the information displayed can be read easily as can be seen in the response to comment 2.

4. Line 129: "In other words, do the associations between frontoparietal neurochemicals and visuomotor processing is replicated…" – a typo here.

We have now corrected the typo, and the text now reads, “In other words, do the associations between frontoparietal neurochemicals and visuomotor processing replicate across different tasks such as numerical processing…” (page: 6, paragraph: 1, line: 5)

5. Line 134: "A more optimal and informative way to address these questions is by using…" This is a strange way to begin a paragraph given that you have not discussed an alternative – less optimal – way of addressing those questions in the previous paragraph. Consider rewriting for clarity.

We have now amended the text, and the Introduction now states, “An informative way to address these questions is by using magnetic resonance spectroscopy (MRS).” (page: 6, paragraph: 2, line: 10)

6. Line 139: "Therefore, it is possible to utilize and extend this unique predictive ability of MRS to discern which of these key neurobiological processes (cognitive, decision, visuomotor) are shaped by highly specialized neurochemical concentration across development." – This language seems to imply that finding a correlation between MRS detected metabolite concentrations and task performance is indicative of a causal relationship. I think it's important to be clear that correlations, even those that are predictive of future states, are not necessarily indicative of causal relationships.

We absolutely agree. The text now reads, “Therefore, it is possible to utilise and extend this unique predictive ability of MRS to discern which of these key neurobiological processes (cognitive, decision, visuomotor) are associated with highly specialised neurochemical concentrations across development (Porges et al., 2021).” (page: 6, paragraph: 2, line: 13)

7. Supplementary File 2: This file could be made clearer. Including more than two plots on each page, and grouping them according to either task or model parameters would allow for easier comparison. Some of the y-axis labels have typos, e.g., DISTANCEFARMINUSCLOSE, which I think are probably due to the way MATLAB treats underscores in text. Perhaps try using different clearer or correct these issues in a separate figure editing package. I think it would be useful to include an explanation of all the terms (e.g., alerting, orienting, executive, etc.) at the top of the file.A large number of correlations were tested in this study. I appreciate that the authors performed an FDR correction and tested the test-retest reliability with a 1.5-year gap. I think that it would also be useful to see the distribution of all the correlation values, i.e., in a histogram, to get an idea of how distinguishable those that were categorized as meaningful are from those that were categorized as not meaningful.I understand the choice of colour schemes for the scatterplots in the main text (blue and red for excitation and inhibition, respectively), however, the individual datapoints are difficult to parse between groups because they are all the same colour and semi-transparent. To improve the clarity of the plots, you could use different colours for each group, or something similar. Also, making the dots smaller will reduce the overlap and probably make them easier to distinguish from one another.

We have now given clearer variable names as indicated by the title of each Supplementary File. Moreover, we have arranged the figures to allow a more straightforward comparison.

Regarding the second point, we plotted a histogram of the correlation values. Still, we doubt this alone is informative in distinguishing those meaningful from those not.

**Author response image 1. sa2fig1:** 

Instead, we now present the statistical results of each of the correlations in Supplementary File 2, including the correlation coefficient, the confidence intervals and the adjusted R^2^ of the linear, quadratic and cubic fits, and we have also compared the three fits to identify which one was the most superior. To do that, we first generated 5000 bootstrapped samples for each of the three fits, and then we performed a two-sample t-test comparing the adjusted R^2^ for each pair to designate which fit was superior. We also rank (based on the uncorrected Pearson P-value) the correlations based on the FDR value to meaningfully distinguish those that stand out as significant from the others.Regarding the latter point, we have implemented the reviewer’s suggestion to make the dots smaller to reduce the overlap and make it easier to distinguish from one another. We have also improved the clarity of the plots by using different colours for each group.

8. Figures 2-4: Please indicate the units of the x and y labels. Also, please indicate in the figure caption whether higher scores in visuomotor processing are associated with better or worse task performance.

We now provided these pieces of information for the x and y labels. We also indicate in the figure caption this additional information and the text states, “Lower scores in processing indicate better performance.”.

9. Figures 2-4: "For visualization purposes, we did not control for boundary separation and mean drift rate when plotting these panels." – I don't understand how not controlling for the variability helps visualization, could the authors please clarify?The authors find a correlation between IPS GABA and Glu and visuomotor processing in three different tasks. It would be interesting to test whether there is independent variability explained by the neurotransmitters in each task or not. For example, how correlated are the visuomotor processing scores between tasks, and if you control for the variability in one/two tasks, is there still a correlation between GABA/Glu and visuomotor processing in the third task?

First, we revised the abovementioned sentence to make it more clear to “To make the visual comparison across figures more comparable, we did not control for boundary separation and mean drift rate when plotting these panels.”. Moreover, the text now reads, “We additionally plotted the moderating role of age in the relation between neurotransmitter concentration and visuomotor processing after controlling for boundary and drift rate in Figure 2—figure supplement 1.” (page: 11, paragraph: 1, line: 1).

Second, as the reviewer suggested, we correlated the three parameters across tasks (see table below), and we identified (and report now in Supplementary File 10) positive correlations of the three EZ parameters across tasks even though the EZ parameters were quantified in different cognitive tasks and using a different number of trials. This supporting evidence suggests that the current study’s EZ parameters were reliably estimated.

Third, even though our study did not focus on the task-specific EZ parameters, we nonetheless conducted the analyses suggested by the reviewer, which we present in Author response table 2. As can be seen, controlling for the EZ parameter of the other two tasks renders the neurotranmistter*age interaction no longer significant.

**Author response table 2. sa2table2:** 

	DF	β	T	CI_L	CI_U	P value
Exp1 IPS MeanDriftRate	223	0.012292	0.251672	-0.0731	0.0965	0.788716
Exp1 IPS glutamate BoundarySeparation	221	-0.02617	-0.60067	-0.1256	0.0694	0.611227
Exp1 IPS glutamate NonDecisionTime	218	-0.08893	-2.21777	-0.2108	0.0407	0.166475
Exp2 IPS glutamate MeanDriftRate	223	0.045953	1.069436	-0.0455	0.1327	0.316705
Exp2 IPS glutamate BoundarySeparation	220	-0.05899	-1.25181	-0.1886	0.0705	0.378585
Exp2 IPS glutamate NonDecisionTime	217	-0.07223	-2.06421	-0.1608	0.0178	0.112948
Exp3 IPS glutamate MeanDriftRate	222	0.02705	0.587266	-0.0512	0.1152	0.534832
Exp3 IPS glutamate BoundarySeparation	219	0.034256	0.626466	-0.0935	0.1566	0.603508
Exp3 IPS glutamate NonDecisionTime	216	-0.14387	-2.54742	-0.3079	0.035	0.099783
Exp1 IPS GABA MeanDriftRate	224	-0.00284	-0.05867	-0.0989	0.0833	0.955603
Exp1 IPS GABA BoundarySeparation	221	0.019515	0.458234	-0.0483	0.1022	0.623926
Exp1 IPS GABA NonDecisionTime	218	0.09984	2.522437	-0.0058	0.1951	0.050963
Exp2 IPS GABA MeanDriftRate	224	-0.0121	-0.28504	-0.0843	0.0683	0.768743
Exp2 IPS GABA BoundarySeparation	221	0.003692	0.077367	-0.1085	0.1183	0.953672
Exp2 IPS GABA NonDecisionTime	218	0.093119	2.767306	0.0057	0.169	0.025182
Exp3 IPS GABA MeanDriftRate	223	-0.03727	-0.82348	-0.1272	0.0674	0.461595
Exp3 IPS GABA BoundarySeparation	221	0.124429	2.382694	0.0053	0.2423	0.039292
Exp3 IPS GABA NonDecisionTime	218	0.056507	0.915952	-0.1465	0.2371	0.575368

10. Line 367: "These findings suggest that the relationship between IPS glutamate and GABA and visuomotor processing across development is task-independent." – This is related to the previous comment. This may have very little to do with GABA/Glu and more to do with the reliability of measuring the same latent construct with these three tasks. I think that the analysis suggested above could help to test this and to put this assertion into a better context.A recent study found evidence for a brain-wide positive relationship between GABA and Glu concentration in adults (Rideaux et al., 2022, NeuroImage). Thus, one might predict that if GABA were positively correlated with task performance, then Glu would also be positively correlated. By contrast, in the current study it appears that for the majority of correlations between GABA and task performance, there appears to be an inverse relationship with Glu. This seems quite interesting, could the authors speak to this? For example, does the relationship between GABA and Glu change with age?

Yes, in our sample, the GABA and glutamate change with age. We have also reported that in the manuscript:

“Even though not clearly established yet, the age trajectories of GABA and glutamate during childhood and early adulthood have been examined in several studies to date. For example, there is evidence for increasing GABA and macromolecules during childhood (Porges et al., 2021), but this has been ascribed to macromolecules rather than GABA itself (Bell et al., 2021). In a separate investigation using the same sample we have found increases in GABA and decreases in glutamate from early childhood to early adulthood (for extensive Discussion on the age trajectories of MRS-based metabolites literature see (Zacharopoulos, Emir, et al., 2021)).” (page: 22, paragraph: 3, line: 22)

We have also checked whether the relationship between glutamate and GABA within the IPS is moderated by age. To do that, we predicted glutamate concentration from three predictors (i) GABA concentration, (ii) age and (iii) the GABA concentration*age interaction. Moreover, we predicted GABA concentration from three predictors (i) glutamate concentration, (ii) age and (iii) the glutamate concentration*age interaction. None of the two metabolite*age interaction terms was significant.

11. The authors found relationships between IPS GABA/Glu and task performance, but not MFG GABA/Glu. This could be due to poorer signal-to-noise in the MFG measurements. Can the authors test this possibility by comparing measures of signal quality between the voxel locations, i.e., FWHM, CRLB, SNR, and fitting residuals?

We have now run these additional analyses. MFG (vs IPS) exhibited lower SNR (T(272)=-19.4898, P<0.0001, CI=[-10.0704 , -8.2226]), higher line width (T(274)=8.0696, P<0.0001, CI=[0.0044 , 0.0072]), higher GABA CRLB (T(275)=11.2841, P<0.0001, CI=[6.1168 , 8.7021]) and higher glutamate CRLB (T(275)=2.5673, P=0.0108, CI=[0.1310 , 0.9922]), suggesting a poorer signal in the MFG. However, the data from both regions used in this study were scrutinised against a set of well-defined exclusion criteria, and we only used the ones deemed reliable. Therefore we believe that the data from both regions have a good level of signal-to-noise ratio. Moreover, in a separate investigation using different measures and a subset of the sample used in the current study, we found robust effects for the neurochemical concentration within the MFG but not IPS (Zacharopoulos, Sella, and Cohen Kadosh, 2021). Taken together, this set of evidence suggests that our neurochemical measures from both regions are overall robust enough to detect the effects of interest.

12. Line 310: "Since we identified a strong and task-independent developmental effect between IPS neurochemicals and visuomotor processing…" – By what categorization method is this effect strong? Most of the β values seem relatively weak (e.g., between 0.15-0.20). Perhaps the authors mean reliable?

We appolgise for this inaccuracy. The text now reads, “Since we identified a reliable task-independent developmental effect between IPS neurochemicals and visuomotor processing…” (page: 17, paragraph: 3, line: 23)

13. Line 447: "…through the visuomotor resting-state connectivity depending on the individual's developmental stage." – This claim relies on both positive results, e.g., evidence of a relationship at some ages, and negative results, e.g., evidence of no relationship at other ages. However, the latter can only be established by directly testing the likelihood of the null hypothesis, e.g., with Bayesian statistics.

We have now removed that claim to accurately reflect the empirical data, and the text now reads, “Here we propose a generic, resting-state mechanistic model by which the levels of excitation and inhibition within the left IPS shape task-independent visuomotor functions through the visuomotor resting-state connectivity across development. However, several aspects of this proposed model are still not known. For example, (i) are there specific time windows during visuomotor processing when the levels of IPS excitation and inhibition regulate the activity of visuomotor regions? Do the levels of IPS excitation and inhibition regulate the activity of visuomotor regions by regulating local GABA and glutamate within visuomotor regions? Addressing these questions is beyond the scope of the present study but can be examined in an investigation combining functional MRS and task-based fMRI (Jia et al., 2022; Koolschijn et al., 2021).” (page: 26, paragraph: 2, line: 14)

14. The paragraph starting at line 461: Related to the above point, several explanations are provided for why the relationship was only found for older participants. Another explanation is that the data may have been noisier for younger participants which led to a failure to detect the relationship.

In our study, we used a motion scrubbing threshold was 2mm, and we excluded cases where more than 5% of the scans were removed, and the number of cases removed for each age group was as follows: early childhood=29, late childhood=4, early adolescence=1, later adolescence=3, early adulthood=0. Even though more data from younger participants were excluded (as expected), the data from all participants used in this study were scrutinised against a set of well-defined exclusion criteria, and we only used the ones deemed reliable. Therefore we believe that the data from all age groups exhibit a good level of signal-to-noise ratio. Furthermore, in a separate investigation using different measures, we found robust effects in the case of early childhood compared to later developmental (Zacharopoulos et al., 2022). Taken together, this set of evidence suggests that our neurochemical measures are overall robust enough to detect effects in several developmental groups, even in the case of early childhood.

15. Please provide a plot showing the positioning of the voxels, ideally with the degree of overlap between participants.

We now provide those plots in Figure 1—figure supplement 2. The positions displayed in a T1-weighted image for the (A) middle frontal gyrus (MFG) and the (B) intraparietal sulcus (IPS), are shown on sagittal and axial slices, respectively. The mean MNI coordinates (SD in parentheses) were as follows, MFG: x=-28.3031 (4.5408), y=31.7434 (11.9630), z=20.5135 (9.5048) and for the IPS: x=-27.4906 (4.1632), y=-56.1867 (8.4567), z=41.2382 (7.2391).

16. Please report the following MRS information: spectral width, number of data points acquired, and number of water-unsuppressed averages.

The text now reads, “The number of spectral data points for the MFG and the IPS was 2048, and the spectral bandwidth was 6000 Hz.”. (page: 35, paragraph: 2, line: 11)

17. Please report the phase and frequency alignment method used on subspectra prior to averaging.

We now report this and the next now reads, “The MRspa pre-processing steps included the following: (1) cross-correlation method to minimise the frequency difference between single shot MRS data, (2) least square method to minimise the phase difference between single shot MRS data, (3) Eddy Current Correction (ECC) was used to correct for both ECC based on the phase of water reference scan and (4) save the averaged spectrum.” (page: 31, paragraph: 2, line: 21)

18. Please report the test-retest reliability of the GABA and Glu concentration estimates and compare them with previously reported test-retest reliability studies, e.g., Saleh et al., 2016, Magn. Reson. Mater. Phys. Biol. Med.It is reported in the Methods that metabolite estimates were rejected as outliers prior to performing tissue correction. I think that tissue correction should be performed prior to outlier rejection.

In the study mentioned by the reviewer, the authors (Saleh et al., 2016) examined the reproducibility of GABA measurements across different software (i.e., LCModel, jMRUI and GANNET) in 20 participants. They found that reproducibility (within-subject coefficient of variation), ranged from 13-22% (anterior cingulate) and 13-18% (parietal). In that study, the authors examined raw (GABA_H2O_) and GABA-to-Creatine ratio (GABA/Cr). The current study assessed test-retest reliability for 3T using the approach described elsewhere (Terpstra et al., 2016). Since we utilised arguably one of the most established methods in the literature since the early 2000s including the LCModel analysis (Tkáč et al., 2001), we believe that our methods represent reliable tools for neurochemical analyses. The text now reads, “Overall, we utilised one of the most established methods in the literature since the early 2000s including the LCModel analysis (Tkáč et al., 2001).” (page: 32, paragraph: 3, line: 19)

Moreover, in our sample, the neurochemical scores were positively related between the first and the second assessment, both in the case of glutamate (P<.001) and GABA (P<.001) as expected.

Regarding the latter point raised by the reviewer, some exclusion criteria were applied only before tissue correction, and some were applied only after tissue correction. For example, if the SNR of a given case (output by the lcmodel) was deemed removable (given our SNR-based exclusion criterion), then this case was excluded before tissue correction because it was problematic from the start. However, other exclusion criteria, for instance, if a case fell beyond 3SD per tissue-corrected metabolite score, then this case would have been excluded AFTER the tissue correction because the standard deviation was based on the tissue-corrected metabolite score (not the raw scores).

19. Line 682: "…as this helped to assess the whether the results…" – typo.

Many thanks for noticing that. We have now corrected that and the text now states “as this helped to assess whether the results…” (page: 39, paragraph: 2, line: 24)

20. Line 713: "…but this did not survive the neurochemical and neurotransmitter specificity…" – What is the difference between neurochemical and neurotransmitter specificity?

We have now amended the text which now states, “neuroanatomical and neurotransmitter specificity”. (page: 41, paragraph: 1, line: 7). Neuroanatomical refers to a finding identified for one of the anatomical regions but not the other (e.g., IPS but not MFG) and neurotransmitter specificity refers to a finding identified for one neurotransmitter but not the other (e.g., glutamate but not GABA).”

21. Please report the order in which participants performed the 3 tasks (was it always the same, randomized, or counterbalanced?). Related to this, FDR correct was performed on the attention network task correlations, but not on the subsequent "replication" correlations with the other two tasks (digit comparison and mental rotation). Could the authors please specify whether analyses were performed on the attention network task, prior to analysing the correlations between GABA/Glu and the other two tasks?

We thank the reviewer for these important comments. The attention network task was counterbalanced with other tests (e.g., intelligence scores), but it was always administered before the mental rotation task, which was followed by the digit comparison. The authors can confirm that the authors performed the analysis on the attention network task, chronologically prior to analysing the correlations between GABA/Glu and the other two tasks. As for the preregistration, the reviewer is absolutely right. We did not preregister this study, which was started at 2014, before this culture change occurred.

22. Line 760: "Since we obtained the same pattern of findings regarding visuomotor processing across Task 1-3 which was one of our earlier aims was addressed…" – typo.

We have now amended that, and the text now reads, “Since we obtained the same pattern of findings regarding visuomotor processing across Task 1-3, which was one of our earlier aims,” (page: 43, paragraph: 1, line: 3)

23. Line 764: "zscored" – hyphenate.

We changed zscore to z-scored.

References

Aiken, L. S., West, S. G., and Reno, R. R. (1991). *Multiple regression: Testing and interpreting interactions*. sage.

Barron, H., Vogels, T. P., Emir, U., Makin, T., O’shea, J., Clare, S., Jbabdi, S., Dolan, R. J., and Behrens, T. (2016). Unmasking latent inhibitory connections in human cortex to reveal dormant cortical memories. *Neuron*, *90*(1), 191-203.

Bell, T., Stokoe, M., and Harris, A. D. (2021). Macromolecule suppressed GABA levels show no relationship with age in a pediatric sample. *Scientific reports*, *11*(1), 1-7.

Chuang, M.-T., Liu, Y.-S., Tsai, Y.-S., Chen, Y.-C., and Wang, C.-K. (2016). Differentiating radiation-induced necrosis from recurrent brain tumor using MR perfusion and spectroscopy: a meta-analysis. *PloS one*, *11*(1), e0141438.

Draheim, C., Mashburn, C. A., Martin, J. D., and Engle, R. W. (2019). Reaction time in differential and developmental research: A review and commentary on the problems and alternatives. *Psychological Bulletin*, *145*(5), 508.

Dyke, K., Pépés, S. E., Chen, C., Kim, S., Sigurdsson, H. P., Draper, A., Husain, M., Nachev, P., Gowland, P. A., and Morris, P. G. (2017). Comparing GABA-dependent physiological measures of inhibition with proton magnetic resonance spectroscopy measurement of GABA using ultra-high-field MRI. *Neuroimage*, *152*, 360-370.

Frangou, P., Emir, U. E., Karlaftis, V. M., Nettekoven, C., Hinson, E. L., Larcombe, S., Bridge, H., Stagg, C. J., and Kourtzi, Z. (2019). Learning to optimize perceptual decisions through suppressive interactions in the human brain. *Nature communications*, *10*(1), 1-12.

Hong, D., Rohani Rankouhi, S., Thielen, J.-W., Van Asten, J. J., and Norris, D. G. (2019). A comparison of sLASER and MEGA-sLASER using simultaneous interleaved acquisition for measuring GABA in the human brain at 7T. *PloS one*, *14*(10), e0223702.

Ip, I. B., Emir, U. E., Parker, A. J., Campbell, J., and Bridge, H. (2019). Comparison of neurochemical and BOLD signal contrast response functions in the human visual cortex. *Journal of Neuroscience*, *39*(40), 7968-7975.

Jia, K., Frangou, P., Karlaftis, V. M., Ziminski, J. J., Giorgio, J., Rideaux, R., Zamboni, E., Hodgson, V., Emir, U., and Kourtzi, Z. (2022). Neurochemical and functional interactions for improved perceptual decisions through training. *Journal of neurophysiology*, *127*(4), 900-912.

Joers, J. M., Deelchand, D. K., Lyu, T., Emir, U. E., Hutter, D., Gomez, C. M., Bushara, K. O., Eberly, L. E., and Öz, G. (2018). Neurochemical abnormalities in premanifest and early spinocerebellar ataxias. *Annals of neurology*, *83*(4), 816-829.

Kolasinski, J., Logan, J. P., Hinson, E. L., Manners, D., Zand, A. P. D., Makin, T. R., Emir, U. E., and Stagg, C. J. (2017). A mechanistic link from GABA to cortical architecture and perception. *Current Biology*, *27*(11), 1685-1691. e1683.

Koolschijn, R. S., Shpektor, A., Clarke, W. T., Ip, I. B., Dupret, D., Emir, U. E., and Barron, H. C. (2021). Memory recall involves a transient break in excitatory-inhibitory balance. *eLife*, *10*, e70071.

Logothetis, N. K., Pauls, J., Augath, M., Trinath, T., and Oeltermann, A. (2001). Neurophysiological investigation of the basis of the fMRI signal. *Nature*, *412*(6843), 150-157.

Near, J., Andersson, J., Maron, E., Mekle, R., Gruetter, R., Cowen, P., and Jezzard, P. (2013). Unedited in vivo detection and quantification of γ‐aminobutyric acid in the occipital cortex using short‐TE MRS at 3 T. *NMR in Biomedicine*, *26*(11), 1353-1362.

Porges, E. C., Jensen, G., Foster, B., Edden, R. A., and Puts, N. A. (2021). The trajectory of cortical GABA across the lifespan, an individual participant data meta-analysis of edited MRS studies. *eLife*, *10*, e62575.

Ratcliff, R., Love, J., Thompson, C. A., and Opfer, J. E. (2012). Children are not like older adults: A diffusion model analysis of developmental changes in speeded responses. *Child development*, *83*(1), 367-381.

Saleh, M. G., Near, J., Alhamud, A., Robertson, F., van der Kouwe, A. J., and Meintjes, E. M. (2016). Reproducibility of macromolecule suppressed GABA measurement using motion and shim navigated MEGA-SPECIAL with LCModel, jMRUI and GANNET. *Magnetic Resonance Materials in Physics, Biology and Medicine*, *29*(6), 863-874.

Schmiedek, F., Oberauer, K., Wilhelm, O., Süß, H.-M., and Wittmann, W. W. (2007). Individual differences in components of reaction time distributions and their relations to working memory and intelligence. *Journal of Experimental Psychology: General*, *136*(3), 414.

Schubert, A.-L., and Frischkorn, G. T. (2020). Neurocognitive psychometrics of intelligence: How measurement advancements unveiled the role of mental speed in intelligence differences. *Current directions in psychological science*, *29*(2), 140-146.

Schubert, A.-L., Frischkorn, G. T., Hagemann, D., and Voss, A. (2016). Trait characteristics of diffusion model parameters. *Journal of Intelligence*, *4*(3), 7.

Schulz-Zhecheva, Y., Voelkle, M. C., Beauducel, A., Biscaldi, M., and Klein, C. (2016). Predicting fluid intelligence by components of reaction time distributions from simple choice reaction time tasks. *Journal of Intelligence*, *4*(3), 8.

Terpstra, M., Cheong, I., Lyu, T., Deelchand, D. K., Emir, U. E., Bednařík, P., Eberly, L. E., and Öz, G. (2016). Test‐retest reproducibility of neurochemical profiles with short‐echo, single‐voxel MR spectroscopy at 3T and 7T. *Magnetic resonance in medicine*, *76*(4), 1083-1091.

Tkáč, I., Andersen, P., Adriany, G., Merkle, H., Uǧurbil, K., and Gruetter, R. (2001). in vivo 1H NMR spectroscopy of the human brain at 7 T. *Magnetic Resonance in Medicine: An Official Journal of the International Society for Magnetic Resonance in Medicine*, *46*(3), 451-456.

von Krause, M., Lerche, V., Schubert, A.-L., and Voss, A. (2020). Do Non-Decision Times Mediate the Association between Age and Intelligence across Different Content and Process Domains? *Journal of Intelligence*, *8*(3), 33.

Wagenmakers, E.-J., Van Der Maas, H. L., and Grasman, R. P. (2007). An EZ-diffusion model for response time and accuracy. *Psychonomic bulletin and review*, *14*(1), 3-22.

Zacharopoulos, G., Emir, U., and Cohen Kadosh, R. (2021). The cross‐sectional interplay between neurochemical profile and brain connectivity. *Human brain mapping*, *42*(9), 2722-2733.

Zacharopoulos, G., Sella, F., Cohen Kadosh, K., Emir, U., and Cohen Kadosh, R. (2022). The effect of parietal glutamate/GABA balance on test anxiety levels in early childhood in a cross-sectional and longitudinal study. *Cerebral Cortex*, *32*(15), 3243-3253.

Zacharopoulos, G., Sella, F., Cohen Kadosh, K., Hartwright, C., Emir, U., and Cohen Kadosh, R. (2021). Predicting learning and achievement using GABA and glutamate concentrations in human development. *PLoS biology*, *19*(7), e3001325.

Zacharopoulos, G., Sella, F., and Cohen Kadosh, R. (2021). The impact of a lack of mathematical education on brain development and future attainment. *Proceedings of the National Academy of Sciences*, *118*(24), e2013155118.

Zatorre, R. J., Fields, R. D., and Johansen-Berg, H. (2012). Plasticity in gray and white: neuroimaging changes in brain structure during learning. *Nature neuroscience*, *15*(4), 528-536.

[Editors' note: further revisions were suggested prior to acceptance, as described below.]

The manuscript has been improved but there are some remaining issues that need to be addressed, as outlined below.While the reviewers appreciate the detailed responses to the initial comments, there remain a number of specific issues that you need to address in a revision (see detailed comments from the two reviewers below). In particular, both reviewers think you need to be more open about the limitations of the data and analyses and temper some of the claims. This applies to issues such as correlations between GABA/Glu and task performance (R1), analyses of signal quality (R1), and mediation analyses (R2).Additional analyses and careful revisions will be required for a positive decision on the manuscript.Reviewer #2 (Recommendations for the authors):The revised manuscript is an improvement on the original and the authors have addressed most of my previous reviews. However, there remain a number of outstanding issues, which will need to be addressed.1. There are both red and pink data points in Figures 2-6, yet these don't appear to be labelled or described. Please fix this if it is a visualization error, otherwise please explain the distinction.

To avoid confusion, the figure legend in Figures 2-6 now states, "Color-coded chronological age increases gradationally from red to purple to blue to green. Red colors represent -1 SD below the mean (younger participants), blue colors represent the mean, and green colors represent +1 SD above the mean (older participants)." (page: 10, paragraph: 1, line: 9)

2. The authors misunderstood my previous request. I wasn't asking whether task performance is correlated. I had asked if there were independent relationships between GABA and task performance and Glu and task performance. For example, if you partial out the variability of GABA from Glu, does the relationship between Glu and task performance disappear? If so, this would make it difficult to infer specific associations between GABA/Glu and these tasks. In particular, Glu and GABA measurements tend to share variance, due to both metabolic (e.g., synthesis cycles) and non-metabolic factors (e.g., signal quality, frequency drift, subject age). If task performance is associated with this shared pool of GABA-Glu variance, then it is difficult to say which neurotransmitter is involved or (worse) if the task isn't simply correlated with the variance associated with non-metabolic factors. I think this is an important point and should be addressed.

We would like to thank the reviewer for this suggestion which we have now implemented (which we pasted below). Our analysis indicates that the first option is the correct one, as both predictors were significant when included in the regression model. The text now states: "Additional analyses presented in Supplementary File 15 revealed that there were independent relationships between IPS glutamate*age and IPS GABA*age with task performance, as both predictors (i.e., IPS glutamate*age and IPS GABA*age) explained unique variance in task performance (non-decision time) even when both were added in the same multiple regression model." (page: 9, paragraph: 2, line: 17)

3. The authors found that the signal quality in MFG was significantly poorer than that in IPS, as indicated by multiple indicators of signal quality. Please report this in the main text of the manuscript and acknowledge that this could explain why a relationship was not detected between MFG GABA/Glu and task performance. I appreciate that the authors feel confident that this is not the explanation; however, I think it's important that readers be made aware of the significantly lower signal quality of the MFG spectra.

We now report this in the main manuscript, and the text now reads, "Although the measurements from both the MFG and the IPS yielded excellent spectral quality, due to degraded shimming conditions in MFG, the line width of MFG was slightly higher (MFG=.027, IPS=.022, P<0.001). Due to the excellent spectal quality, we think that these differences are unlikely to explain the lack of effect of MFG GABA and glutamate, but the reader should be aware of such possibility." (page: 23, paragraph: 1, line: 20)

4. The authors indicate that they are confident that the MRS measurements from all age groups have sufficient signal quality to detect meaningful relationships with their additional measures because they have excluded the datasets with poorest signal quality. This point is also offered in R2.R11. Matching the datasets in terms of their minimum signal quality is not sufficient to show that signal quality differences cannot explain the failure to detect relationships with other measurements. To make this claim, the authors need to demonstrate that the average signal quality is similar between groups. Please compare the average signal quality between groups, as was done between IPS and MFG in R2.R11. and report whether there are significant differences in the main text of the manuscript. If the average signal quality is significantly lower in groups where relationships were not detected, please acknowledge that lower signal quality may explain the failure to detect these relationships.

As requested, we have examined the relationship between average signal quality and age, which we now report in the main text, "By performing bivariate correlations, we found that older (vs younger) participants exhibited lower SNR (P<0.001), higher line width (P<0.001), and higher glutamate CRLB (P<0.001) across both regions, which is expected as we have shown previously that older participants exhibit lower T2 values (Zacharopoulos et al., 2021)." (page: 34, paragraph: 1, line: 6).

Moreover, the text now states, “Please note that even though lower signal quality may explain the failure to detect these relationships for certain groups of participants (e.g., younger participants in this instance), as can be seen in Materials and methods, this was not the case as the average signal quality was higher in younger participants (where this relationship was not detected) compared to older participants (where this relationship was detected).” (page: 28, paragraph: 1, line: 9).

**Author response table 3. sa2table3:** 

	**Spearman's Rho**	**P**	**se**	**CI_L**	**CI_U**
IPS_SNR	-0.222	<.001	0.059	-0.334	-0.103
MFG_SNR	-0.329	<.001	0.055	-0.434	-0.215
IPS_lw	0.338	<.001	0.054	0.227	0.439
MFG_lw	0.403	<.001	0.05	0.299	0.495
IPS_GABA_CRLB	-0.12	0.058	0.064	-0.244	0.008
IPS_Glutamate_CRLB	0.506	<.001	0.046	0.413	0.592
MFG_GABA_CRLB	-0.02	0.757	0.064	-0.144	0.105
MFG_Glutamate_CRLB	0.486	<.001	0.047	0.391	0.575

In Author response images 2-8, we plotted line graphs with error bars (95% confidence intervals) for each age group (1 = Younger, 5 = Older).

**Author response image 3. sa2fig3:** 

**Author response image 4. sa2fig4:** 

**Author response image 5. sa2fig5:** 

**Author response image 6. sa2fig6:** 

**Author response image 7. sa2fig7:** 

**Author response image 8. sa2fig8:** 

5. Please increase the luminance of the T1-weighted images to make the structure more visible and provide axial, sagittal, and coronal views for each voxel.

We have now increased the luminance of the T1-weighted images to make the structure more visible and provided axial, sagittal, and coronal views for each voxel in Figure 1—figure supplement 2.

6. I think that the authors have misinterpreted my request – I agree that LCModel is an established method of metabolite quantification. I requested that the authors show the test-retest reliability of the GABA/Glu measurements for two reasons: (1) this is useful comparative information regarding MRS measurements and will be of interest to a broad range of readers and (2) because it provides the reader with an indication of the dynamic reliability/consistency of these measurements, which is influenced by many factors other than the quantification algorithm applied. Thus, could the authors please provide the test-retest reliability of the measurements (not simply the p-values for the correlation between them)?

We thank the reviewer for this clarification. We report the relevant details in the text, stating: "Moreover, as shown previously, our methods produced robust test-retest reliability in a previous study (Terpstra et al., 2016) where glutamate NAA, tCr, tCho, and Ins were quantified with between-session coefficients of variance of ≤5% even at 3T, and GSH mimicking the same GABA condition had a coefficient of variance of ~10%." (page: 33, paragraph: 2, line: 10)

Of note, the overall signal of the current study is expected to be higher than the Terpstra et al. study since the age group used in the Terpstra et al. study was older, and as we showed in the current study age is negatively associated with signal quality. Our data examined the effect of GABA and glutamate over an average of approximately 21 months, rather than at a much shorter scale (minutes to days). Therefore the effect of development does not allow us to address test-retest reliability. However, as seen in the table below, we calculated the between-subject coefficients of variation (CoV) for each age group (G1: Youngest, G5: Oldest), each region, and each metabolite. CoV values are lower than previously published between subject CoVs (Wijtenburg and Knight‐Scott, 2011). In our study, the average CoV was comparable or lower to the average CRLB for GABA and glutamate, suggesting that relatively small variations in concentrations may be detectable at different ages (Van de Bank et al., 2015), which is line with our previous test-retest findings (Terpstra et al., 2016). These findings, together with previous findings in the field, increase the confidence in the reliability of our measure.

This was calculated as follows:

CoV=(standard deviation of metabolite across participants)/(mean of metabolite across participants)*100

**Author response table 4. sa2table4:** 

	G1	G2	G3	G4	G5
IPS GABA	11.48	14.08	16.25	13.43	16.83
IPS glutamate	6.74	7.13	8.34	7.54	8.14
MFG GABA	18.32	17.28	17.05	20.22	18.68
MFG glutamate	6.46	7.25	8.48	8.00	8.33

Reviewer #3 (Recommendations for the authors):In my view, the authors were moderately responsive to reviewer comments. I appreciate the authors' clarifications of prior work on DDM parameters and intelligence, the motivation for the supplemental experiment, and the resting state fMRI analysis procedures. I also think that the addition of the parameter correlation tables and the supplemental EFA increase confidence that the Ter parameter being measured is a task-general construct. I cannot speak as to whether the additional details about the MRS aspects of the study would impact the other reviewers' concerns, but it does seem like there is now greater transparency about the MRS procedures. I still disagree with the choice to investigate DDM parameters across the three tasks as three manifest variables rather than a single latent variable and the choice to visualize effects by binning ages rather than showing age as a continuous factor, but in both cases, I could see why there are reasonable arguments for the choices the authors made. However, I still have several concerns that I do not think were adequately addressed.Simply showing the fits of nonlinear models to age differences in DDM parameter values in supplemental materials does not really address the possibility that non-linear effects of age could impact the main analyses reported in the paper, in which linear age appears to moderate the associations between DDM parameter values and MRS measures. There appears to be clear nonlinearity in relationships between age and Ter, the parameter that received the greatest focus. Unless there is a clear reason why sensitivity analyses that include nonlinear age terms would not be possible, such sensitivity analyses would be very valuable for determining whether the primary findings of the study are robust to these considerations.

To address this possibility, we added the analysis below to Supplementary File 16. We hope that this analysis addresses the reviewer's feedback.

The evidence from the mediation analysis is still concerningly weak. The authors now acknowledge the post hoc nature of the analyses and the liberal, uncorrected α level (although the other reason for concern, the inconsistency of effects across age groups, does not seem to be acknowledged as a limitation of the evidence for these effects). However, the mediation effects are still framed as a central claim of the paper. I think it makes sense to report the mediation results for completeness and to describe them as preliminary findings that could be replicated, but given the weakness of the evidence I don't think it makes sense to describe these results as a central finding. For example, the statement in the abstract that "We showed that fluid intelligence performance is explained by IPS GABA and glutamate and is mediated by visuomotor processing" seems like much too strong of a claim given the available evidence and the inconsistency in the details of the mediation effects across age groups. The age moderation effects of associations between DDM parameters and MRS measures that are reported seem to be robust and interesting in their own right, which makes me question the need to highlight a mediation analysis that provides evidence that, in my view, can be thought of as preliminary at best.

We understand the reviewer's concern. At the same time, we included the fluid intelligence test as this was one of the main aims of our study. To address the reviewer's concern, we toned down the wording in the abstract, stating, "We present evidence that fluid intelligence performance is explained by IPS GABA and glutamate and is mediated by visuomotor processing. However, this evidence was obtained using an uncorrected α and needs to be replicated in future studies.". We hope that this provides more balance and will allow to push the field forward.

I appreciate the authors' addition of key details about how the DDM parameters were estimated. It seems like several of these details would be appropriate to mention as limitations in the discussion, including the relatively low number of trials and the high accuracy rate of the tasks, both of which can negatively impact the reliability of parameter estimates. However, a parameter recovery study would greatly increase confidence in the assumption that these features of the task data did not impede the accurate estimation of DDM parameters.

We have now addressed these issues, and the text reads, "Another limitation of the current study regarding the computation of the mean drift rate, boundary separation and non-decision time stems from the relatively low number of trials (although a larger number of participants) and the high accuracy rate of the tasks used, both of which can negatively impact the reliability of parameter estimates. However, it was previously demonstrated that the EZ ability to accurately reflect individual differences in model parameters (both in large, 800 trials, and in small, 80 trials, data sets) and its ability to recover the experimental effects on parameter means (van Ravenzwaaij and Oberauer, 2009) often outperformed that of other diffusion models such as fast-dm (Voss and Voss, 2007) and DMAT (Vandekerckhove and Tuerlinckx, 2007)." (page: 30, paragraph: 2, line: 4)

References

Terpstra, M., Cheong, I., Lyu, T., Deelchand, D. K., Emir, U. E., Bednařík, P., Eberly, L. E., and Öz, G. (2016). Test‐retest reproducibility of neurochemical profiles with short‐echo, single‐voxel MR spectroscopy at 3T and 7T. *Magnetic resonance in medicine*, *76*(4), 1083-1091.

van Ravenzwaaij, D., and Oberauer, K. (2009). How to use the diffusion model: Parameter recovery of three methods: EZ, fast-dm, and DMAT. *Journal of Mathematical Psychology*, *53*(6), 463-473.

Vandekerckhove, J., and Tuerlinckx, F. (2007). Fitting the Ratcliff diffusion model to experimental data. *Psychonomic bulletin and review*, *14*, 1011-1026.

Voss, A., and Voss, J. (2007). Fast-dm: A free program for efficient diffusion model analysis. *Behavior research methods*, *39*(4), 767-775.

Wijtenburg, S. A., and Knight‐Scott, J. (2011). Very short echo time improves the precision of glutamate detection at 3T in 1H magnetic resonance spectroscopy. *Journal of Magnetic Resonance Imaging*, *34*(3), 645-652.

Zacharopoulos, G., Emir, U., and Cohen Kadosh, R. (2021). The cross‐sectional interplay between neurochemical profile and brain connectivity. *Human brain mapping*, *42*(9), 2722-2733.